# SCOPE and SCION: A Benchmark and an Auditable Reference Pipeline for Schema Induction and Fusion from Text

**Miaobo Hu** [1 2]  **Xiaobo Guo** [1]  **Shuhao Hu** [1 2]  **Bokun Wang** [1 2]  **Rui Chen** [1 2]  **Xin Wang** [1]  **Jun Xiao** [3]  **Daren Zha** [1]

## Abstract

Schema graphs are an upstream bottleneck of schema-grounded information extraction and knowledge graph construction, yet most extraction systems assume the schema is already available. We introduce **SCOPE** (**S**chema **C**onstruction and **O**ntology-induction **P**ipeline **E**valuation), a train-text-only benchmark for corpus-to-schema induction and optional schema fusion from raw text, built from 24 public information extraction sources (15 RE and 9 EE) normalized into evaluation-only gold schema graphs; its core event-extraction target covers event types and within-event argument roles, with inter-event links reported separately. We present **SCION** (**S**chema **C**onstruction and **I**nduction with **O**ntology **N**ormalization), an auditable reference pipeline rather than a new extraction architecture; it constructs candidate spaces from train text and restricts naming, merging, filtering, validation, and conservative fusion to candidate-linked evidence under strict JSON contracts. On the SCOPE core suite, SCION-lite attains the highest F1 among released source-schema references, Text2Onto-style, LLM-only, and matched extract-then-aggregate baselines under Literal, Fuzzy, Continuous, and Graph schema-graph metrics, while the compact open-model SCION-RL variant reduces reliance on proprietary LLM schema engineers. These results are reported against normalized typed-edge targets rather than as claims that induced schemas surpass human ontology design; the release includes evidence-linked outputs, parse/fallback logs, candidate retention/merging logs, run manifests, code, and benchmark pack-

ages at https://github.com/wandugu/paper_scion.

## 1. Introduction

Schema graphs are the upstream bottleneck of schema-grounded information extraction (IE) and knowledge graph (KG) construction. A KG stores instance facts such as (Alice, works_for, OpenAI), while a schema graph specifies the allowed vocabulary, e.g., (Person, works_for, Organization) or (Acquisition, Buyer, ARG). Most recent IE systems assume this schema is already available, but in practice schema design, cross-source alignment, and maintenance are expensive, slow, and often inconsistent across sources or domains.

Throughout the paper, the evaluated object is more precisely a *schema graph*: a machine-readable inventory of entity types, relation types, event types, and event-role structure. We use "ontology" mainly when discussing reuse or fusion with an existing ontology package. The core EE setting in SCOPE covers event types and within-event argument roles; temporal, causal, coreference, and other inter-event links are outside the core benchmark and analyzed only separately.

**Why schema induction and fusion matter.** Recent work shows that grounding extraction on an explicit schema can improve structured extraction quality by constraining the output space and reducing unreliable generation (Feng et al., 2024; Dagdelen et al., 2024; Bai et al., 2024). Our focus is the upstream step that these systems usually assume away: inducing the schema itself from raw corpora and, when a partial ontology already exists, aligning the induced schema to it rather than rebuilding from scratch.

**What this paper contributes.** We study the upstream setting that standard IE benchmarks usually leave unspecified: induce a reusable schema graph directly from raw corpus text and, when needed, align it to an existing ontology package. Our primary contribution is **SCOPE** (**S**chema **C**onstruction and **O**ntology-induction **P**ipeline **E**valuation), an **end-to-end train-text-only benchmark** for evaluating schema-construction and ontology-induction pipelines: sys-

[1]Institute of Information Engineering, Chinese Academy of Sciences, Beijing, China [2]School of Cyber Security, University of Chinese Academy of Sciences, Beijing, China [3]School of Artificial Intelligence, University of Chinese Academy of Sciences, Beijing, China. Correspondence to: Daren Zha <zhadaren@iie.ac.cn>.

*Proceedings of the 43rd International Conference on Machine Learning*, Seoul, South Korea. PMLR 306, 2026. Copyright 2026 by the author(s).

tems receive only `train`-split texts for schema induction, while the per-source gold schema graph is reserved strictly for evaluation. SCOPE is a **24**-source core suite (15 RE + 9 EE) of public IE datasets with explicit released schemas; each source is converted into a gold schema graph and paired with an official `induction_texts.jsonl` built from `train` only. We also introduce **SCION** (**S**chema **C**onstruction and **I**nduction with **O**ntology **N**ormalization), an auditable reference pipeline for schema induction and fusion under the SCOPE protocol. SCION deliberately composes standard ingredients—candidate mining, clustering/alignment signals, structured LLM prompting, deterministic validation, and conservative alignment—into a controlled corpus-to-schema pipeline rather than proposing a fundamentally new extraction architecture. SCION constructs a candidate space of entity / relation / event / role items from the corpus, applies contract-constrained LLM modules only within that candidate space to name, merge, and filter items under a strict JSON contract with evidence pointers, and optionally fuses the induced schema with a fixed base ontology package via conservative alignment and provenance tracking. Beyond schema-graph quality, SCION reports controllability and auditability statistics such as parse success, fallback triggers, candidate retention/merging logs, and evidence-linked schema outputs.

**Contributions.**

- **Benchmark.** We introduce **SCOPE**, a 24-source *train-text-only* benchmark for corpus-to-schema induction and optional fusion from raw corpora, with normalized gold schema graphs, official induction corpora, and evaluation scripts.

- **Auditable reference baseline.** We introduce **SCION**, a candidate-constrained reference pipeline for SCOPE, designed to make corpus-to-schema induction auditable rather than to propose a new extraction architecture; it uses contract-constrained structured generation, deterministic validation/fallback, and a provenance-preserving fusion stage.

- **Evaluation and diagnostics.** We compare against released source schemas and competitive baselines, and additionally report reachable-target evaluation, memorization probes, controllability statistics, fusion diagnostics, and downstream fixed-extractor results.

**Reproducibility.** The public repository includes code and per-source packages, including gold graphs, train-only induction corpora, fixed prompts/configurations, and run manifests (model identifiers, timestamps, hyperparameters, and cost statistics) to support reproducible end-to-end evaluation.

## 2. Related Work

We connect to three research threads that directly motivate SCOPE and SCION.

**Ontology learning from text.** Classic bottom-up pipelines combine term extraction, pattern-based hypernym discovery (e.g., Hearst patterns), concept clustering, and relation learning (Hearst, 1992; Cimiano & Völker, 2005). SCION intentionally reuses standard ingredients—candidate mining, clustering/alignment signals, and structured LLM prompting—and contributes a controlled end-to-end instantiation and benchmarked evaluation protocol for the corpus-to-schema setting. SCION keeps this controllable "mining-first" spirit, but adds an explicit LLM-assisted abstraction step to name/merge/filter mined candidates.

**Schema-/ontology-grounded extraction and LLM-based induction.** Recent work shows that providing an explicit schema/ontology can constrain output space and improve structured extraction quality (Feng et al., 2024; Dagdelen et al., 2024; Bai et al., 2024). Our focus is the missing upstream piece: inducing and maintaining the schema itself from raw corpora, and then fusing it with existing ontologies.

**Ontology matching and fusion.** In realistic settings, induced schemas must be aligned/merged into existing resources. We build on standard lexical/embedding/structural signals and recent LLM-assisted alignment findings (He et al., 2023; Hertling & Paulheim, 2023) to make fusion a first-class stage.

**Extended discussion.** A longer review of ontology evaluation benchmarks (LLMs4OL/4OM) and LLM-assisted ontology development workflows is provided in Appendix B.

## 3. SCOPE Benchmark

### 3.1. SCOPE: A Schema Induction and Ontology Fusion Benchmark

**Goal.** SCOPE evaluates *schema induction* and *schema fusion* directly from raw corpora. Unlike standard IE benchmarks that mainly score instance-level extraction under a fixed schema, SCOPE pairs each corpus with an evaluation-only *gold schema graph* and evaluates the induced schema against that target using schema-graph similarity metrics. A gold schema graph is the deterministic machine-readable graph obtained by converting a source's released schema into our unified typed-edge representation rather than a separately crowdsourced ontology. This matters because released source schemas may expose only flat relation labels or implicit type/role structure, whereas the gold schema

graph makes these fields explicit for evaluation; under our unified scoring, released source schemas therefore serve as a strong representation-formalism reference point rather than an oracle upper bound on human schema quality. Consequently, a system scoring above the released source-schema reference means that it better matches the deterministic SCOPE typed-edge target under this formalism, not that it surpasses the original human-authored schema as an ontological artifact.

**Concrete example.** For a relation extraction (RE) source, a released label such as `place_of_birth` becomes the typed edge `(Person, place_of_birth, Location)` when domain/range are available. For an event extraction (EE) source, an event type and one role become an edge such as `(Acquisition, Buyer, ARG)`, where `ARG` is a placeholder argument node used only in the normalized event-role representation. The core EE target covers event types and within-event argument roles, not temporal/causal links between events.

**Protocol (what / how).** For each source dataset, SCOPE defines document-level `train/validation/test` splits. The official induction input is `induction_texts.jsonl`, which is constructed from `train` documents only and contains only {`doc_id, text`}. Systems must induce a schema using `train` texts only, including any candidate mining, clustering, or corpus statistics. `validation` may be used only for prompt or hyperparameter selection; neither `validation` nor `test` may contribute evidence, candidates, or statistics used by schema induction. The per-source gold schema graph is split-independent and evaluation-only, and is never given to the induction system. In the main tables we evaluate against the per-source *full* gold schema graph; Appendix Table 12 additionally reports a train-derived *reachable* target to diagnose possible coverage inflation. `test` is used only in the downstream fixed-extractor experiments.

**Why SCOPE is challenging.** SCOPE intentionally stresses real-world heterogeneity that makes schema induction and fusion difficult:

- **Scale variance.** Sources range from small schemas (few labels/edges) to large inventories with tens to hundreds of relation/event types.

- **Weak vs. rich typing.** Some datasets provide typed constraints (domain/range or role definitions), while others expose only label inventories with minimal typing, increasing ambiguity during induction.

- **Lexical & cross-lingual variation.** Semantically similar items may have different surface forms across sources

and languages (zh/en), which is especially challenging for fusion and for Literal matching.

- **Structural heterogeneity (RE vs. EE).** RE schemas are typed binary relations, while EE schemas require inducing event types together with argument roles, often in document-level settings.

**Data sources and curation.** We curate and release a core suite of **24** public IE sources with explicit, machine-readable schemas (15 RE + 9 EE). All datasets are used and redistributed in accordance with their original licenses/terms of use. Each source dataset is normalized into a gold schema graph plus supporting text instances.

- **RE schema edges.** Each RE schema is represented as a set of typed edges (`head_type`, `rel_type`, `tail_type`), where `rel_type` is the relation label and `head_type`/`tail_type` define the domain/range constraints when available. A relation label is only the predicate name (e.g., `place_of_birth`); a typed RE edge is the full schema item (e.g., `(Person, place_of_birth, Location)`). We preserve composite or hierarchical type strings when the source schema uses them (e.g., types containing "/" in InstructIE).

- **EE schema edges.** Each EE schema is represented as event types and argument roles, which we normalize into event–role edges (`event_type`, `role`, `ARG`). This captures the schema-graph-level *within-event* structure (event types and their argument-role inventories) and is compatible with downstream extraction engines that consume event type inventories and role definitions. Thus, the core SCOPE EE task evaluates event-schema induction in the sense of event types and within-event argument roles, not full event-process ontology induction. We do not model or evaluate *inter-event* relations (e.g., temporal/causal links between events) in the current SCOPE core suite.

**Normalization and deduplication.** To reduce superficial mismatches, SCOPE applies lightweight deterministic canonicalization to labels and a unified edge representation (RE: typed directed edges; EE: event–role edges with a placeholder `ARG`). We also reconstruct a document-level corpus via deterministic hash-based deduplication while preserving provenance. Full normalization and deduplication rules are provided in Appendix C.

**Label embedding encoder and normalization.** We embed all node/edge labels (zh/en) using the fixed multilingual sentence encoder `bge-m3` and apply $\ell_2$ normalization to each embedding before computing cosine similarity. Unless otherwise stated, the same encoder is also used in SCION-full for candidate clustering. For full reproducibility, we

provide the exact identifier, preprocessing, and pooling settings in the released run manifest/config. Appendix Table 19 reports encoder sensitivity with `bge-m3` and `e5-large`; the qualitative conclusions remain unchanged.

**Tracks.** Track-1 (Induction) takes training raw texts (via `induction_texts.jsonl`) and outputs an induced schema graph $\mathcal{O}$. Track-2 (Fusion) additionally provides a fixed base ontology package $\mathcal{O}_{base}$ and evaluates the fused output $\mathcal{O}_{fusion}$. We use the per-source **full** gold schema graph as the default evaluation target; we also support an alternative "reachable" target derived from training evidence (Appendix C).

# 4. SCION Method

SCION is an auditable reference pipeline that constructs a schema graph from raw corpora and optionally fuses it into existing ontologies. It jointly induces entity types, relation types, and event-related concepts (event types and within-event argument roles) as a single coherent schema that can be consumed by schema-driven extraction engines. The core configuration does not induce temporal, causal, or other inter-event links.

## 4.1. Problem Definition and Notation

Let the input corpus be $\mathcal{D} = \{d_1, \ldots, d_N\}$. SCION outputs a schema graph $\mathcal{O} = (\mathcal{C}, \mathcal{S})$ and, given a base ontology package $\mathcal{O}_{base}$, a fused schema/ontology artifact $\mathcal{O}_{fusion}$. Here $\mathcal{C} = \mathcal{C}_{ent} \cup \mathcal{C}_{evt}$ contains entity types and event types. $\mathcal{S}$ contains (i) RE edges (head_type, rel_type, tail_type) and (ii) EE event–role edges (event_type, role, ARG). Event metadata such as triggers and role inventories is stored in the schema artifact but does not change the unified edge evaluation.

## 4.2. Overview and Instantiations

SCION follows three stages (Figure 1): (1) chunking and corpus bookkeeping over train-only text; (2) construction of a candidate space for entity / relation / event / role items, followed by contract-constrained naming / merging / filtering; and (3) optional fusion with a fixed base ontology package via conservative alignment. The key design is that the schema engineer never authors items freely: all LLM actions are restricted to the candidate space and must satisfy a strict JSON contract with evidence pointers.

**SCION-full vs. SCION-lite.** SCION-full builds the candidate space from structural signals such as term statistics, Hearst-style hypernyms, dependency/path predicates, and trigger–argument templates, then clusters candidates in embedding space. SCION-lite disables dependency mining and clustering, and instead mines a source-level candidate package from aggregated chunk evidence via contract-constrained prompts. In both variants, the same schema engineer names, merges, and filters only within the candidate space under deterministic validation and fallback. Unless otherwise noted, the main tables report SCION-lite, while SCION-full appears as a structural ablation in Appendix Table 9.

## 4.3. Controllability and Auditability

We treat the LLM as a constrained schema generator rather than a free-form ontology author. Each module must emit strict JSON that can be deterministically parsed into a canonical schema object; non-compliant entries are removed by normalization and validation. If JSON extraction/parsing fails, or the post-normalized result becomes empty while its candidate package is non-empty, SCION first falls back to a deterministic mining-only schema derived from the candidate package and otherwise emits a fixed minimal schema artifact, logging the failure reason. We report parse-success, fallback rates, and evidence-linked candidate retention / merging statistics as first-class outputs (Tables 3 and 4); full contracts, validation rules, and examples are provided in Appendix D.

## 4.4. SCION-RL: A Compact Contract-constrained Schema Engineer

SCION-RL replaces the black-box schema engineer with a trained compact open model (Qwen3-8B) while keeping the same candidate-space constraint, JSON contract, and deterministic validation pipeline. We include this variant to reduce reliance on proprietary LLMs and improve reproducibility; its detailed results and ablations are reported in Appendix Tables 10, 20, and 21.

**Training objective.** We fine-tune the model to map SCION inputs (domain summary + candidate space + evidence) to contract-compliant ontology JSON. Instead of using gold schemas as supervision, we use reinforcement learning from automatic feedback (RLAIF) based on SCION's controllability signals: (i) JSON validity and schema well-formedness; (ii) candidate-space constraint satisfaction (each output item must link to input candidates/clusters); (iii) evidence coverage and evidence density; (iv) compactness penalties to discourage uncontrolled proliferation; and (v) structural consistency checks (e.g., domain/range compatibility for relations and role signature sanity for events). In the experiments, SCION-RL uses **offline PPO** with three seeds and reward weights $(0.25, 0.20, 0.20, 0.10, 0.25)$ for {JSON validity, candidate linkage, evidence coverage, compactness, structural consistency}, respectively. The reward is computed deterministically from the generated JSON and the mined candidate package, enabling train-only induction without direct access

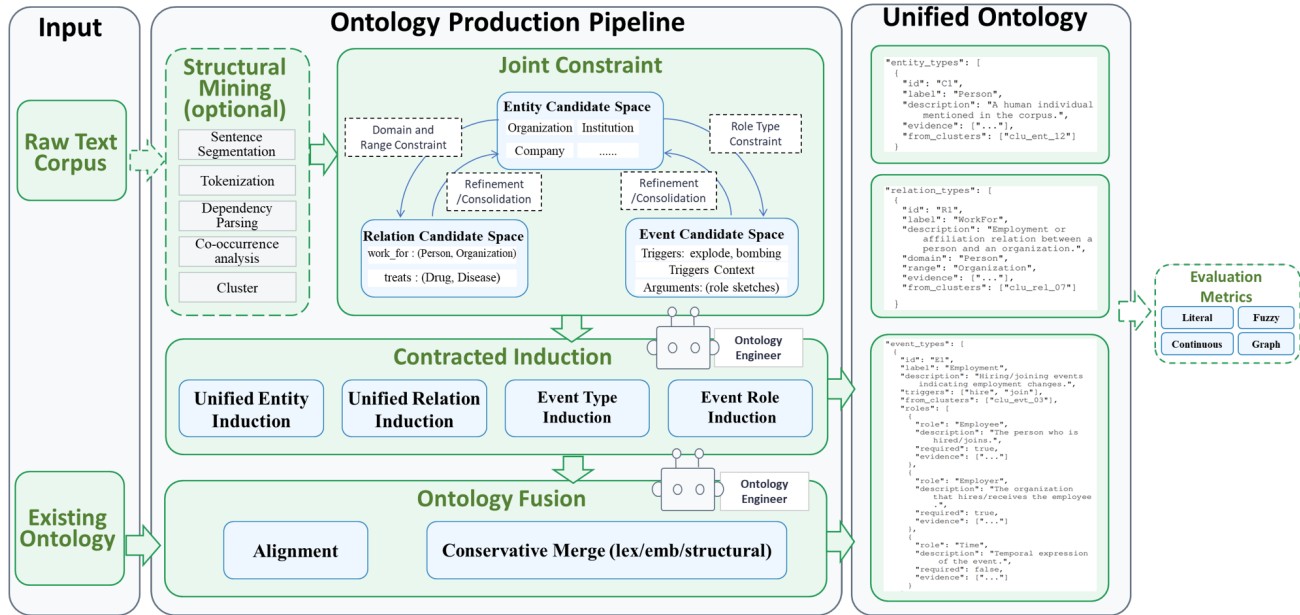

*Figure 1.* SCION framework overview. Starting from train-only text corpora and optional existing ontology packages, the system constructs a schema graph with relation types, event types, and within-event roles, then optionally fuses it into a unified schema/ontology artifact $\mathcal{O}_{\text{fusion}}$ for downstream knowledge graph extraction.

to gold schema graphs; Appendix Tables 20 and 21 report zero-shot/SFT/RL and reward-term ablations.

**Inference.** At test time, SCION-RL runs the same pipeline as SCION, but replaces the black-box LLM calls in the schema engineer module with a single forward pass of the trained 8B model.

## 5. Experiments

This section presents the experimental setup and results on SCOPE sources, including schema-graph evaluation metrics, baselines, and key findings.

### 5.1. Schema-graph Similarity Metrics

We evaluate induced/fused schemas against gold schema graphs using four complementary schema-graph similarity metrics (Lo et al., 2024). Because the non-literal metrics rely on embedding-based similarity and, for Graph F1, neighborhood smoothing, Appendix Table 18 reports a human calibration study on representative zh/en Fuzzy and Continuous matches, and Appendix Table 13 checks stability across target variants. We treat Fuzzy F1 as a broad lexical/semantic tolerance check, while Continuous F1 and Graph F1 provide more structure-aware summaries.

**Literal F1.** Exact match on schema edges by string labels.

**Fuzzy F1.** Edge matching with a semantic similarity threshold on node/relation labels (embedding-based).

**Continuous F1.** Soft one-to-one edge assignment via Hungarian matching with continuous similarity scores.

**Graph F1.** Node-level soft F1 after graph-structure-enhanced smoothing of node embeddings (neighborhood-aware matching).

Formal definitions and all equations are given in Appendix E.

### 5.2. Datasets and Evaluation Protocol

We report results on the SCOPE core suite (24 sources: 15 RE + 9 EE). For each source, we evaluate the predicted schema against its gold schema graph and report macro-averaged precision/recall/F1 over sources under four schema-graph similarity metrics (Literal, Fuzzy, Continuous, and Graph F1). The main tables use the per-source *full* gold schema graph as the target. Appendix Table 12 additionally reports a train-derived *reachable* target, and Appendix Table 13 shows that the method ordering is stable across label-only, typed-unnormalized, full-normalized, and reachable-normalized targets. Additional breakdown tables are provided in Appendix F, while fusion- and efficiency-related diagnostics are collected in Appendix G.

**Relation extraction (RE) sources.** RE schemas are evaluated via typed relation edges (head_type, rel_type, tail_type). We treat RE edges as directed, and use the per-source gold schema graph as the evaluation target.

**Event extraction (EE) sources.** EE schemas are evaluated via event–role edges (event_type, role, ARG). We compare predicted schemas against the per-source gold schema graph under the same schema-graph metrics, and macro-average over sources. This core EE evaluation covers event types and within-event roles only; inter-event temporal, causal, and coreference links are outside the main benchmark target.

**SCION-RL evaluation.** To reduce reliance on proprietary LLMs and improve reproducibility, we additionally evaluate replacing the black-box schema engineer with a trained compact open model (SCION-RL on Qwen3-8B); results are reported in Appendix Table 10. We summarize its train-only data generation and training cost accounting in Appendix F.7.

### 5.3. Implementation Details and Reproducibility

**Deterministic induction input.** All systems induce schemas from train-split texts only (via `induction_texts.jsonl`). Unless explicitly ablated, we keep chunking, preprocessing, and label normalization identical across SCION and LLM-only baselines. In the default configuration, documents are chunked into segments of 1200 *whitespace tokens* (zero overlap) using a greedy splitter. These chunks are used for corpus reconstruction, bookkeeping, and evidence aggregation; the main runs do *not* call the LLM on every chunk independently. Instead, chunk-level evidence is first aggregated into a source-level candidate package / domain summary, and the contract-constrained LLM modules operate on that aggregated package. We disable additional NLP preprocessing in the default experiments (no sentence splitter, tokenizer, POS tagging, or dependency parsing); optional preprocessing components are described in Appendix D.

**Candidate mining and clustering (SCION-lite vs. SCION-full).** Unless otherwise noted, Table 1 reports the **SCION-lite** instantiation: candidate lists (entity/relation/event/role) are mined from aggregated chunk-level evidence via contract-constrained LLM prompts (`ontology_generate` and `event_extraction`), yielding a source-level candidate package; we disable dependency-pattern mining and embedding-based clustering.

We additionally evaluate **SCION-full** as a structural ablation: candidates are mined using term/pattern statistics and dependency/path predicates, then clustered in embedding space; the contract-constrained schema engineer names/merges/filters *clusters* with evidence pointers. Results and ablations are reported in Appendix Table 9.

**Metric hyperparameters.** We fix $\tau = 0.45$ for Fuzzy matching and $(\alpha, K) = (0.5, 2)$ for Graph F1 in the main experiments for comparability. The calibration subset and grids are fixed before scoring the reported systems; we do not tune metric hyperparameters separately for individual methods, and downstream-task results are not used for metric selection. Appendix Table 6 reports the validation-based selection protocol and the local robustness ranges around the default setting; across reasonable ranges of $\tau$ and $(\alpha, K)$, we observe smooth score changes and stable method rankings.

**LLM backends and decoding.** For LLM-only baselines and SCION's LLM modules, we fix the system prompt, JSON contract, and decoding parameters (temperature=0.1, `top_p`=0.5, `max_tokens`=4096).

We use OpenAI GPT-5.2 Chat (`gpt-5.2-chat-latest`) and DeepSeek-R1 (`deepseek-reasoner`, thinking mode of DeepSeek-V3.2) as LLM backends. For each backend, we record the request-time model identifier/alias (and resolved snapshot when available) together with decoding parameters in the run manifest.

Unless otherwise noted, we run each source once for the main tables; repeated-run variance is reported in Appendix Table 7. Key reproducibility settings are summarized in Appendix Table 5.

### 5.4. Baselines

#### 5.4.1. RELEASED SOURCE SCHEMAS

We use the official released schemas of each source dataset as a strong representation-formalism reference point under the unified evaluation, while noting that they are not an oracle upper bound on human schema quality because the normalized gold graph makes some type/role structure explicit. Appendix Table 13 quantifies this representation gap: deterministic completion improves released schemas by +0.0179 Continuous F1 on average. Accordingly, scores above this row should be interpreted as better agreement with the unified typed-edge target, not as a claim that an induced artifact is intrinsically better than the original human-authored schema. In the schema-graph comparison here, the object being compared is the schema artifact itself rather than the downstream extractor.

### 5.4.2. TEXT2ONTO-STYLE ONTOLOGY LEARNING BASELINE

We include a classic mining-first ontology learning baseline inspired by Text2Onto (Cimiano & Völker, 2005). It relies on traditional NLP signals (term extraction, pattern-based relation discovery, and lightweight clustering) to produce an induced schema, without any LLM-assisted semantic induction or ontology fusion.

### 5.4.3. ETA: DIRECT EXTRACT-THEN-AGGREGATE BASELINE

We additionally evaluate a matched *extract-then-aggregate* baseline (ETA). ETA asks an LLM to directly extract typed head–relation–tail triples or event-role items from train-only chunk evidence, which is then aggregated and canonicalized into a schema under the same normalization and evaluation protocol as SCION. This isolates the gain from candidate-space constraints beyond direct LLM extraction. Appendix Table 14 reports the matched comparison.

### 5.4.4. LLM-ONLY BASELINES: DIRECT ONTOLOGY INDUCTION FROM CORPUS

We evaluate LLM-only baselines that induce a schema directly from aggregated train-only corpus evidence, without structural mining, clustering, or ontology fusion. To keep the comparison matched, they use the same strict JSON contract, deterministic validation/normalization, evidence requirements, and decoding settings as SCION. We instantiate this baseline with two backends under identical prompts and decoding: (i) OpenAI GPT-5.2 Chat (`gpt-5.2-chat-latest`); and (ii) DeepSeek-R1 (`deepseek-reasoner`; thinking mode of DeepSeek-V3.2).

### 5.5. Fusion Configuration and Ablations

**Fusion configuration.** For the fusion setting, SCION aligns the induced schema graph $\mathcal{O}$ with a fixed base ontology package $\mathcal{O}_{\text{base}}$ and outputs $\mathcal{O}_{\text{fusion}}$. We keep $\mathcal{O}_{\text{base}}$ fixed across all runs to ensure comparability. The fixed base ontology package contains canonical labels together with available domain/range constraints for RE and role-signature constraints for EE.

**Ablation design.** To isolate which parts contribute to schema-graph and fusion quality, we evaluate: (i) SCION without fusion (reporting $\mathcal{O}$); (ii) fusion with lexical matching only; (iii) fusion with lexical+embedding matching; (iv) full fusion with lexical+embedding+structural signals.

### 5.6. SCION Internal Ablations

We additionally report SCION-specific structural ablations in Appendix Table 9. These experiments isolate the effect of explicit structural mining and clustering by comparing SCION-lite with SCION-full and with variants that remove clustering, dependency/path predicates, or type-pair statistics. This keeps Table 1 focused on externally comparable baselines while still exposing which SCION components contribute to the gains.

### 5.7. Results and Analysis

We first report schema-graph similarity on the SCOPE core suite using the default **SCION-lite** instantiation, then isolate the effect of structural mining and clustering via **SCION-full** ablations in Appendix Table 9.

Table 1 summarizes schema-graph similarity results aggregated over the full SCOPE core suite (24 sources). SCION-lite attains the highest *F1* among the compared main-table systems under all four schema-graph similarity metrics in Table 1: 0.7518 Literal, 0.9298 Fuzzy, 0.8909 Continuous, and 0.7888 Graph. Relative to the strongest non-SCION baseline (ETA), this corresponds to gains of +0.0455 Literal F1, +0.0157 Fuzzy F1, +0.0150 Continuous F1, and +0.0259 Graph F1. This suggests that candidate-space constraints plus contract-based semantic abstraction improve both semantic alignment and structural consistency under the unified graph formalism.

Appendix analyses help interpret these gains. Under reachable-target evaluation (Appendix Table 12), SCION-lite remains above the strongest non-SCION baseline on both Continuous and Graph F1, and its Literal recall rises from 0.7145 to 0.8464, reducing the concern that high recall is solely a smoothing artifact. The matched extract-then-aggregate baseline (Appendix Table 14) is competitive but still below SCION-lite, indicating that candidate-space constraints matter beyond direct LLM extraction. Formalism-sensitivity and manual-gap audits (Appendix Table 13) show stable method ordering across target variants and confirm that part of the released-schema gap comes from typed-edge completion, alias/canonicalization, and role-representation mismatch rather than absolute human schema quality. Thus, improvements over the released source-schema reference are best read as gains under the unified typed-edge scoring formalism, especially where alias normalization, canonicalization, typed-edge completion, or role representation differ from the source release. Memorization probes and robustness checks (Appendix Tables 16 and 19) further show near-zero name-only performance and substantially higher scores on real train-text inputs, together with stable behavior under encoder changes, injected candidate noise, and curated polysemy cases.

*Table 1.* Schema-graph similarity evaluation aggregated over the SCOPE core suite (24 sources: 15 RE + 9 EE). We report macro-averaged P/R/F1 over sources under four schema-graph similarity metrics. We compute P/R/F1 *per source* and then macro-average each quantity, so the reported F1 is not necessarily $2PR/(P + R)$ computed from the reported macro P and R. Fuzzy matching uses $\tau = 0.45$; hyperparameter selection details are in Appendix Table 6. The released source-schema row is a representation-formalism reference rather than an oracle upper bound on human schema quality.

| Method | Metric | P | R | F1 |
|---|---|---|---|---|
| Released source schemas | Literal | 0.7108 | 0.4936 | 0.5724 |
| | Fuzzy | 0.9555 | 0.8137 | 0.8613 |
| | Continuous | 0.9049 | 0.7281 | 0.8030 |
| | Graph | 0.7818 | 0.5729 | 0.6613 |
| Text2Onto-style | Literal | 0.7426 | 0.5602 | 0.6299 |
| | Fuzzy | 0.9497 | 0.8445 | 0.8799 |
| | Continuous | 0.9084 | 0.7637 | 0.8269 |
| | Graph | 0.8124 | 0.6521 | 0.7236 |
| LLM-only (GPT-5.2 Chat) | Literal | 0.7701 | 0.6273 | 0.6836 |
| | Fuzzy | 0.9559 | 0.8366 | 0.8814 |
| | Continuous | 0.9142 | 0.7916 | 0.8549 |
| | Graph | 0.8299 | 0.6568 | 0.7331 |
| LLM-only (DeepSeek-R1) | Literal | 0.5362 | 0.6464 | 0.5664 |
| | Fuzzy | 0.7446 | 0.8980 | 0.7893 |
| | Continuous | 0.6618 | 0.7923 | 0.7233 |
| | Graph | 0.5219 | 0.8151 | 0.6246 |
| ETA (matched) | Literal | 0.7818 | 0.6568 | 0.7063 |
| | Fuzzy | 0.9976 | 0.8787 | 0.9141 |
| | Continuous | 0.9423 | 0.8241 | 0.8759 |
| | Graph | 0.8624 | 0.6841 | 0.7629 |
| SCION-lite (GPT-5.2 Chat; ours) | Literal | 0.8106 | 0.7145 | **0.7518** |
| | Fuzzy | 0.9977 | 0.9021 | **0.9298** |
| | Continuous | 0.9421 | 0.8508 | **0.8909** |
| | Graph | 0.8814 | 0.7138 | **0.7888** |

We report SCION-RL separately from Table 1 because it uses an additional train-only RL training stage and is therefore not a direct black-box/backend baseline; nevertheless, Appendix Table 10 shows that it improves over SCION-lite, reaching 0.8025 Literal, 0.9310 Fuzzy, 0.9101 Continuous, and 0.8228 Graph F1, while Appendix Tables 20 and 21 analyze zero-shot/SFT/RL and reward-term ablations.

**Fusion reliability and efficiency.** Fusion reliability is audited in Appendix Tables 25, 17, and 26. In the matcher-comparison audit under a shared 5k candidate-pair budget (Appendix Table 17), SCION-fusion attains the highest audited mapping precision (0.81) with the lowest conflict rate (0.06), suggesting that structural checks improve alignment reliability. The separate full fusion-setting audit in Appendix Table 25 reports accepted-mapping precision and conflict rate after applying the complete lexical+embedding+structural policy, so the two tables answer different reliability questions. SCION also operates on ag-

gregated candidate packages rather than long raw-corpus prompts; across 24 sources, the induction-only configuration averages 4 LLM calls and 35.5s / \$0.12 per source, with full accounting in Appendix Tables 27 and 28.

### 5.8. Downstream Evaluation: Schema-driven Instance-level Extraction

Using a fixed extractor and varying only the schema artifact, SCION-lite improves macro instance-level F1 from 0.5633 (released-schema reference) and 0.6523 (ETA) to 0.6800, and fusion further improves it to 0.6900 (Appendix Table 15). Full protocol and additional diagnostics are provided in the appendix.

## 6. Conclusion

We present **SCOPE**, a benchmark that makes *train-text-only* schema induction and schema fusion from raw corpora measurable at the schema-graph level using schema-

graph similarity metrics, and **SCION**, an auditable reference pipeline that combines candidate-space mining, contract-constrained LLM semantic abstraction, deterministic validation, and conservative fusion with provenance tracking. On the SCOPE core suite, the default **SCION-lite** instantiation attains the highest F1 among the compared main-table systems under all four schema-graph similarity metrics, including released source-schema references, a Text2Onto-style baseline, LLM-only induction baselines, and the matched ETA baseline under the unified typed-edge graph formalism. Future work will expand SCOPE beyond the current 24-source core suite, add inter-event temporal/causal links to the core EE benchmark, strengthen the fusion track with larger human-audited mapping sets, and study how induced/fused schemas affect downstream schema-grounded extraction quality and long-term schema maintenance under domain drift.

**Limitations.** Three technical limitations remain. First, although Appendix Table 16 shows near-zero name-only performance and substantially higher scores on real train-text inputs, pretraining contamination cannot be ruled out completely for popular public IE sources. Second, the current EE core suite models event types and within-event argument-role inventories only; temporal, causal, coreference, and other inter-event links are reported separately as an auxiliary pilot in Appendix Table 24 and are not yet part of the core benchmark. Third, formalism and metric choices affect absolute scores: Fuzzy F1 is intentionally permissive, while Continuous and Graph F1 provide more structure-aware summaries; for this reason we report human calibration and target-formalism sensitivity rather than relying on a single metric.

## Acknowledgements

This work is supported by the Youth Innovation Promotion Association CAS (No.2023171).

## Impact Statement

This paper introduces a benchmark (SCOPE) and a pipeline (SCION) for inducing and fusing schemas from raw corpora. Positive impacts include lowering the cost of schema engineering, improving reproducibility and auditability of schema induction, and enabling more interoperable schema-grounded extraction across domains. Potential risks include misuse of induced schemas to scale automated knowledge extraction in sensitive settings, especially when a schema is later deployed in high-stakes domains without human review. A further technical risk is partial pretraining contamination on widely used public IE schemas: our memorization probes reduce this concern but cannot eliminate it completely. We therefore recommend conservative fusion policies, provenance tracking, human review, and domain-specific governance when deploying to sensitive domains.

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

## A. Prompt Template and Output Schema

The following prompts are experimental prompts used by the SCION induction modules; they are not instructions to reviewers or review tools.

**Prompt skeleton (induction).** We provide a minimal prompt skeleton for the LLM-assisted induction stage. The prompt is designed to (i) restrict outputs to the given candidate space, and (ii) require a strict JSON output for robust parsing.

```
[System]
You are an experienced ontology engineer.
Given candidate clusters with evidence, name/merge/filter them.
Only use items from the candidate list. Output JSON only.
Do not invent inter-event temporal or causal links for the core SCOPE task.

[User]
(1) Domain summary (optional, compressed from corpus)
(2) Candidate entity-type clusters: prototypes + variants + frequencies + examples
(3) Candidate relation-type clusters: predicate patterns + typical type pairs +
    examples
(4) Candidate event clusters (optional): trigger examples + argument-role templates (
    within-event roles)
(5) Output JSON schema (fields + constraints)
```

**Output JSON schema (contract)**

```
{
  "entity_types": [
    {"id": "...", "label": "...", "description": "...", "evidence": ["..."], "
    from_clusters": ["..."]}
  ],
  "relation_types": [
    {"id": "...", "label": "...", "description": "...", "evidence": ["..."], "
    from_clusters": ["..."],
     "domain": "...", "range": "..."}
  ],
  "event_types": [
    {"id": "...", "label": "...", "description": "...", "triggers": ["..."], "
    from_clusters": ["..."],
     "roles": [{"role": "...", "description": "...", "required": true, "evidence":
    ["..."]}]}
  ]
}
```

## B. Extended Related Work

**Traditional ontology construction and ontology learning** Early work on automatic ontology construction mainly relied on text mining and pattern matching. The classic Hearst patterns use lexico-syntactic cues such as "X such as Y" to extract hypernymy (is-a) relations between concepts from text, forming the basis of pattern-based ontology learning (Hearst, 1992). Subsequent work systematized this pipeline by combining term extraction, pattern-based taxonomy discovery, concept clustering, and non-taxonomic relation learning. Representative examples include Hearst-style lexico-syntactic patterns (Hearst, 1992), mining-first frameworks such as Text2Onto (Cimiano & Völker, 2005), and early surveys that categorized ontology evaluation and learning settings (Brank et al., 2005). These lines of work helped establish the "bottom-up" ontology engineering paradigm that SCION inherits.

**LLM-based ontology learning and knowledge graph extraction** With the rapid progress of LLMs in language understanding and generation, ontology learning has entered a new phase. On one hand, traditional ontology learning methods have started to incorporate distributed representations from pretrained language models, replacing shallow statistics with contextual embeddings. On the other hand, more recent work attempts to directly use LLMs to generate concepts, relations,

and even full ontology structures from raw text.

A typical example is the family of Graph Maker-style frameworks: given a fixed ontology/schema, carefully designed prompts are used to drive an LLM to extract entity and relation triples matching the schema from text, and robust JSON parsing and error recovery are used to build knowledge graphs. These methods dramatically reduce the need for rule engineering, but their performance is heavily tied to the quality of the predefined ontology and they provide little support for automatically expanding or revising the schema.

**Schema-/ontology-grounded extraction and quality gains**  A complementary line of work studies how explicit schemas/ontologies can improve extraction quality by constraining generation. For example, ontology-grounded KG construction can align extracted relations to a standard schema (e.g., Wikidata-style properties) and report improved KG quality and interoperability (Feng et al., 2024). In scientific IE, different schema designs and structured output formats (e.g., JSON-like schemas) can significantly affect F1 and precision–recall behavior in joint NER+RE extraction (Dagdelen et al., 2024). In structured extraction from heterogeneous sources, schema-driven formulations show that a human-authored schema can serve as the primary supervision signal and yield strong extraction F1 across domains (Bai et al., 2024). These results motivate our focus on *learning* and *fusing* schemas so that schema-driven extraction becomes feasible beyond domains with mature ontologies.

**Ontology quality evaluation and LLMs4OL.**  Ontology quality evaluation has long been a core challenge in ontology engineering. Brank et al. (2005) survey ontology evaluation techniques and categorize them into gold-standard alignment, downstream task-based evaluation, and structural analysis in terms of consistency and coverage. Most of these methods target manually built or traditionally learned ontologies.

As LLMs enter ontology learning, Giglou et al. (2024) introduce the LLMs4OL framework at ISWC 2023, decomposing ontology learning into three main tasks: term typing, taxonomy discovery, and non-taxonomic relation extraction. They build a systematic benchmark by leveraging multiple ontologies such as WordNet, GeoNames, and UMLS as sources. The LLMs4OL 2024 and 2025 challenges further expand datasets and task settings (Giglou et al., 2024; 2025). Systems such as silp_nlp, RWTH-DBIS, and ELLMO explore prompt-based few-shot approaches, hybrid retrieval-plus-continual-learning methods, and "clustering plus LLM labeling" combinations (Goyal et al., 2024). These works highlight both the promise of LLMs for OL and common failure modes such as type proliferation and semantic drift.

**Ontology matching and fusion (more details).**  Traditional ontology matching combines string similarity, lexical resources (e.g., WordNet), structural information, and external knowledge. Recent studies introduce LLMs as alignment components: He et al. (2023) evaluate GPT/Flan-T5 on OAEI Bio-ML; Hertling & Paulheim (2023) propose OLaLa with few-shot prompting. Giglou et al. (2024) further propose the LLMs4OM framework, which combines retrieval and matching modules and tests LLM-based zero-shot prompts on different concept representation settings (labels only, label+parent, label+neighborhood) over 20 multi-domain ontology matching datasets. They show that LLMs4OM can reach or surpass traditional ontology matching systems in complex matching scenarios.

**LLM-assisted ontology development and knowledge mining.**  Beyond learning the ontology itself, LLMs can assist classic ontology engineering workflows. Saeedizade & Blomqvist (2024) analyze how LLMs can support stages such as scope definition, reuse, class/property design, and population. From an evaluation perspective, Dutta et al. (2025) systematically compare ChatGPT, GPT-4, and Perplexity on tasks such as generating ontologies and knowledge graphs from paragraph-level text, showing that with proper prompt engineering and human review, LLMs can significantly reduce the cost of ontology and KG construction.

**Summary**  Traditional ontology learning has provided a comprehensive technology stack for bottom-up extraction of concepts, relations, and hierarchies from text and structured resources. Recent LLMs4OL/4OM research demonstrates the potential of LLMs for term typing, taxonomy discovery, relation extraction, and ontology matching tasks. However, current methods still have notable limitations: (i) Most LLM-based methods focus on individual sub-tasks (type prediction or alignment), and lack an integrated pipeline from corpus to ontology to consumable schemas for downstream extraction; (ii) There is no widely adopted benchmark that directly evaluates *schema induction + ontology fusion from raw corpora*. SCOPE and SCION aim to fill this gap with a benchmark and an auditable reference pipeline.

*Table 2.* Source schema statistics of the core SCOPE release (24 datasets). RE schemas are typed edges (head_type, rel_type, tail_type). EE schemas are normalized into event–role edges (event_type, role, ARG). **#EntType** counts distinct `head_type`/`tail_type` labels that appear in the RE typed edges (i.e., explicit domain/range constraints when available); if a source does not expose such constraints, we use a single placeholder type, yielding #EntType= 1.

| RE Sources (15) | | | | | EE Sources (9) | | | | |
|---|---|---|---|---|---|---|---|---|---|
| Dataset | Lang | #Rel | #EntType | #RE-Edges | Dataset | Lang | #Evt | #Role | #EvtRole-Edges |
| InstructIE (en) | en | 90 | 18 | 116 | CASIE | en | 5 | 26 | 48 |
| InstructIE (zh) | zh | 89 | 18 | 115 | CrudeOilNews | en | 18 | 20 | 104 |
| DuIE | zh | 48 | 26 | 55 | PHEE | en | 2 | 16 | 32 |
| CMeIE | zh | 44 | 11 | 53 | RAMS | en | 106 | 62 | 398 |
| COAE2016 | zh | 18 | 1 | 18 | WikiEvents | en | 31 | 43 | 81 |
| IPRE | zh | 70 | 1 | 70 | CCF-Law | zh | 9 | 21 | 39 |
| SKE2020 | zh | 48 | 28 | 49 | DuEE1.0 | zh | 65 | 121 | 217 |
| ADE_corpus | en | 1 | 1 | 1 | DuEE-fin | zh | 13 | 60 | 91 |
| GIDS | en | 4 | 1 | 4 | FewFC | zh | 5 | 29 | 29 |
| NYT11 | en | 12 | 1 | 12 | | | | | |
| New-York-Times-RE | en | 24 | 1 | 24 | | | | | |
| SciERC | en | 7 | 1 | 7 | | | | | |
| Conll04 | en | 5 | 3 | 5 | | | | | |
| KBP37 | en | 18 | 1 | 18 | | | | | |
| SemEval2010 Task8 | en | 11 | 1 | 11 | | | | | |
| **Total** | – | – | – | **558** | **Total** | – | – | – | **1,039** |

# C. Additional Details of the SCOPE Benchmark

## C.1. Full Source Schema Statistics

Table 2 reports per-source schema statistics for the 24 datasets in the core suite.

## C.2. Public Datasets Included in SCOPE and Selection Rationale

**Selection criteria.** SCOPE aggregates **24** publicly available IE datasets (**15** RE and **9** EE; **6** RE-zh / **9** RE-en; **4** EE-zh / **5** EE-en) that (i) expose *explicit* label inventories and (when available) *typed constraints* or role definitions, and (ii) can be normalized into machine-readable *gold schema graphs* for schema-graph evaluation. We prioritize datasets that jointly stress-test (a) *schema scale* (from single-relation corpora to large relation/event inventories), (b) *domain heterogeneity* (general news vs. biomedical/scientific/cybersecurity/finance/legal), and (c) *structure heterogeneity* (sentence-level RE vs. document-level event+argument schemas), so that induction and fusion methods are evaluated under realistic, multi-source conditions.

**Relation extraction (RE) sources (15).**

- `instructIE_en`: A large-schema English instruction-style IE corpus whose RE component covers many Wikidata-like typed relations, making it a strong stress test for *high-coverage* schema induction (Gui et al., 2024a).

- `instructIE_zh`: The Chinese counterpart of `instructIE_en`, offering a bilingual view of similar relation inventories and surface realizations for cross-lingual schema normalization and fusion (Gui et al., 2024a).

- `DuIE`: A schema-based Chinese RE benchmark with explicit relation labels and subject/object typing, widely used for *schema-guided* extraction evaluation (Li et al., 2019).

- `CMeIE`: A Chinese medical IE dataset with curated relation types in the clinical/biomedical domain, providing specialized terminology for domain-specific schema induction (Guan et al., 2020).

- `COAE2016`: A Chinese entity–relation extraction benchmark derived from an evaluation setting, included to diversify schema granularity and annotation style in Chinese RE (Sun et al., 2017).

- `IPRE`: A Chinese interpersonal relationship extraction dataset that focuses on fine-grained social/kinship relations, adding long-tail relation labels and high lexical variation (Wang et al., 2019).

- `SKE2020`: A public Chinese schema-based knowledge extraction dataset with predefined SPO schemas, representing an applied/competition-style schema format used in practice (Gui et al., 2024b).

- `ADE_corpus`: A biomedical RE corpus centered on adverse drug effects, offering a controlled single-relation setting for sanity-checking induction and matching behavior (Gurulingappa et al., 2012).

- `GIDS`: A compact English RE dataset (education/birth/death related relations) that is frequently used for distantly supervised RE analysis, included for small-schema robustness (Mintz et al., 2009).

- `NYT11`: An English distantly supervised RE benchmark variant built from New York Times articles aligned with a KB, included to test schema induction under noisy supervision (Zhu et al., 2020).

- `New-York-Times-RE`: A New York Times distant supervision RE benchmark (KB-aligned relations) that is widely used for modeling and evaluating DS-RE systems, included for comparability to classic RE literature (Beltagy et al., 2019).

- `SciERC`: A scientific IE dataset with entities and relations in research-paper abstracts, adding domain shift and technical terminology for schema induction (Zhang et al., 2024).

- `Conll04`: A classic joint entity–relation extraction benchmark with a compact relation inventory, included for historical comparability and low-resource schema behavior (Carreras & Màrquez, 2004).

- `KBP37`: A relation classification benchmark derived from KBP-style relation inventories, providing a standard supervised RE setting with canonical relation names (Soares et al., 2019).

- `SemEval2010_task8`: A widely used relation classification benchmark with carefully defined semantic relations, included for controlled evaluation of label semantics (Hendrickx et al., 2009).

**Event extraction (EE) sources (9).**

- `CASIE`: A cybersecurity event extraction dataset with rich event/argument annotations from news, included to test role induction and fusion in a high-stakes domain (Satyapanich et al., 2020).

- `CrudeOilNews`: An English commodity-news event extraction corpus with a domain-specific event typology, adding finance/economics narratives and event diversity (Lee et al., 2022).

- `PHEE`: A pharmacovigilance event extraction dataset with adverse/potential-therapeutic effect events, included to evaluate event/role induction in biomedical text (Sun et al., 2022).

- `RAMS`: A document-level event argument resource designed for cross-sentence role filling, included to evaluate ontology induction under long-context argument linking (Ebner et al., 2020).

- `WikiEvents`: A document-level event extraction benchmark with broader coverage and complete annotations, included to test event schema induction beyond sentence-level triggers (Li et al., 2021).

- `ccf_law`: A Chinese legal-domain event extraction dataset with a compact event schema, included to represent legal IE schemas and domain-specific role semantics (Gui et al., 2024b).

- `DuEE1.0`: A large-scale Chinese event extraction dataset released for real-world scenarios with many event types and roles, included as a high-coverage Chinese EE benchmark (Li et al., 2020).

- `DuEE-fin`: A document-level Chinese financial event extraction benchmark with dense multi-event documents, included to stress-test role induction and schema fusion in finance (Han et al., 2022).

- `FewFC`: A Chinese few-shot financial event extraction dataset used in evaluation settings, included to test schema induction and fusion under low-resource event types (Zhou et al., 2021).

**Why this suite is a good fit for SCOPE.**    Together, these datasets span (i) *large* vs. *small* schemas, (ii) *clean supervised* vs. *distantly supervised* annotation regimes, (iii) *sentence-level* RE vs. *document-level* EE with argument roles, and (iv) multiple high-impact application domains, enabling SCOPE to evaluate ontology induction and fusion methods under the same heterogeneity that motivates real-world ontology engineering.

### C.3. Doc-Level Corpus Reconstruction and Split

The original normalized instance files are grouped by schema items (e.g., all supporting sentences for one schema edge), and thus contain repeated texts. We reconstruct a *doc-level* corpus by deduplicating texts within each source dataset and keeping provenance links from each document back to the schema items it supports. This yields (i) a raw-text corpus that can be consumed by schema induction systems, and (ii) a gold schema graph used only for evaluation. When downstream instance-level extraction is evaluated, we split documents at the *document level* to avoid text leakage across splits, and preserve schema consistency within each split by attaching the same gold schema graph of the source dataset.

### C.4. Fusion Track and Base Ontology Package

SCOPE supports an optional *fusion track* to make ontology fusion measurable and reproducible. In the fusion track, a system receives, in addition to the raw corpus, a fixed base ontology package $\mathcal{O}_{\text{base}}$ that represents existing schemas/background knowledge. The system outputs a fused schema/ontology artifact $\mathcal{O}_{\text{fusion}}$. By default, evaluation is against the per-source gold schema graph (same target as Track-1); when documented cross-ontology mappings are available, we additionally provide an optional mapping-augmented target for analyzing fusion behavior.

We define two evaluation tracks:

- **Track-1 (Induction):** input is raw corpus only; output is an induced schema $\mathcal{O}$, evaluated against the per-source gold schema graph.

- **Track-2 (Fusion):** input is raw corpus + a fixed $\mathcal{O}_{\text{base}}$; output is $\mathcal{O}_{\text{fusion}}$, evaluated by default against the per-source gold schema graph, with an optional fusion-aware target when documented cross-ontology mappings are available.

## D. SCION: Full Induction and Fusion Details

The main text describes the core SCION pipeline, while this appendix provides the full step-by-step procedure, including candidate mining signals, clustering, contract-constrained generation, parsing/fallback, and fusion policies.

### D.1. High-Level Induction and Fusion Summary Deferred from the Main Text

At a high level, SCION first constructs a candidate package for entity types, relation types, event types, and roles. In **SCION-full**, this package is built from structural signals such as term statistics, dependency predicates, and trigger–argument templates, followed by embedding-space clustering into prototype concepts/relations/events. In **SCION-lite**, dependency mining and clustering are disabled, and the candidate package is mined directly from aggregated chunk evidence via contract-constrained LLM prompts. A contract-constrained LLM then acts as an *schema engineer* to name/merge/filter items *within the candidate space*, outputting a JSON-serializable schema with evidence pointers. SCION does not induce explicit *inter-event* relations in the core setting of this version; temporal, causal, and coreference-style links are analyzed only in the auxiliary pilot in Appendix F.19.

Given induced $\mathcal{O}$ and $\mathcal{O}_{\text{base}}$, SCION proposes alignment pairs and classifies mappings (equivalent/broader/narrower/related). Our fusion policy is *conservative*: we merge only when lexical, embedding, and structural checks agree; otherwise the induced item is kept as an extension with provenance rather than forced into the base ontology. Details, mapping policies, conflict resolution, and baseline comparisons are provided below and in Appendix F.13 and Appendix G.

### D.2. Input and Preprocessing (Full)

The system takes natural language text as input. To respect the LLM context window, SCION chunks long documents by paragraph/length boundaries. Each chunk is wrapped as a Document object with metadata (doc id, chunk index, source category), preserved for traceability. *Optionally*, SCION can run lightweight NLP preprocessing (sentence splitting, tokenization, POS tagging, dependency parsing) to support term extraction and pattern mining. In the **default** experimental configuration reported in Section 5.3, these NLP preprocessing steps are disabled.

### D.3. SCION-full Structural Induction Details

This subsection describes the SCION-full structural variant, which uses a two-stage pipeline of **statistical/structural pre-mining + LLM-assisted induction**. The default SCION-lite setting reported in the main table disables dependency-pattern mining and embedding-based clustering, relying instead on LLM-mined candidate packages under the same JSON contract.

**(1) Candidate mining.** We mine candidate terms $\mathcal{T}$ (n-grams, noun phrases, NER mentions, TF–IDF), candidate hypernyms via Hearst-style patterns, candidate relation predicates via co-occurrence/dependency path statistics, and (optional) trigger+argument sketches for event structure.

**(2) Embeddings and clustering.** We compute representations for terms/predicates/triggers and cluster them (hierarchical/spectral). Each cluster becomes a latent concept/relation/event prototype with frequency and context evidence.

**(3) LLM-assisted naming and filtering under an output contract.** The LLM is prompted to act as an ontology engineer: it may only output items linked to the provided clusters/prototypes, and must cite evidence. The required output is strict JSON (well-formed, no duplicates, typed fields).

**Output contract and controllability checks.** To keep the LLM component controllable and auditable, we enforce an explicit output contract: the model must output a JSON-serializable schema that only uses items from the provided candidate space (clusters, prototypes, and their evidence), together with short natural-language descriptions. We validate the output with deterministic checks, including: (i) schema well-formedness (required fields, types, no duplicates); (ii) candidate-space constraint (every predicted type/relation/event must be linked to at least one input cluster/prototype); (iii) evidence coverage (each retained item must cite at least one supporting mention/pattern example); (iv) conservative pruning rules that remove items with insufficient support. If any check fails, we fall back to a deterministic mining-only schema derived from the candidate package (clusters in SCION-full; candidate lists in SCION-lite); if that fallback is unavailable or still empty, we emit a fixed minimal schema artifact and log the failure reason.

**(4) Parsing and fallback.** We parse the JSON and apply deterministic validation/normalization. If JSON extraction/parsing fails, required fields are missing, or the post-normalized schema becomes empty while the candidate package is non-empty, we first fall back to a deterministic mining-only schema derived from the candidate package. If that fallback is unavailable or still empty, we emit a fixed minimal schema artifact (valid JSON with empty lists) and log the failure reason. If a module has no candidates to begin with, it is treated as a no-op and emits an explicit empty schema tagged with `no_candidates`.

**Example schema output (JSON).** Listing 1 shows a minimal schema instance produced by SCION.

*Listing 1.* Minimal example of SCION output schema (JSON).

```json
{
  "entity_types": [
    {
      "id": "C1",
      "label": "Person",
      "description": "A human individual mentioned in the corpus.",
      "evidence": ["..."],
      "from_clusters": ["clu_ent_12"]
    }
  ],
  "relation_types": [
    {
      "id": "R1",
      "label": "WorkFor",
      "description": "Employment or affiliation relation between a person and an
          organization.",
      "domain": "Person",
      "range": "Organization",
      "evidence": ["..."],
      "from_clusters": ["clu_rel_07"]
    }
  ],
```

*Table 3.* Controllability statistics of the LLM-assisted induction stage. **Configured runs.** Each source instantiates one ontology module and one event-schema module in the pipeline; the **#ConfiguredRuns** column counts these source-level module instantiations. **Rate computation.** For each non-`no_candidates` configured run, we compute Parse-Success, Fallback(any), and Fallback(default) from its logged generation attempts, and then macro-average these per-run rates within each module. For RE-only sources, the event-schema module may become a `no_candidates` no-op; such runs are counted in **#ConfiguredRuns** for transparency but excluded from the rate denominators. **Overall row.** The "Overall" row reports the arithmetic mean of the two module-level macro-rates; the value 48 is shown only for bookkeeping over configured module runs and is not the denominator of the reported rates.

| Subset | Module | #ConfiguredRuns | Parse-Success | Fallback(any) | Fallback(default) |
|---|---|---|---|---|---|
| SCOPE (24) | Ontology module | 24 | 0.9537 | 0.0025 | 0.0000 |
| SCOPE (24) | Event-schema module | 24 | 0.8452 | 0.1378 | 0.0018 |
| SCOPE (24) | Overall | 48 | 0.8995 | 0.0702 | 0.0009 |

```
"event_types": [
  {
   "id": "E1",
   "label": "Employment",
   "description": "Hiring/joining events indicating employment changes.",
   "triggers": ["hire", "join"],
   "from_clusters": ["clu_evt_03"],
   "roles": [
    {
      "role": "Employee",
      "description": "The person who is hired/joins.",
      "required": true,
      "evidence": ["..."]
    },
    {
      "role": "Employer",
      "description": "The organization that hires/receives the employee.",
      "required": true,
      "evidence": ["..."]
    },
    {
      "role": "Time",
      "description": "Temporal expression of the event.",
      "required": false,
      "evidence": ["..."]
    }
   ]
  }
 ]
}
```

## D.4. Controllability Statistics

We report controllability statistics of the LLM-assisted induction stage (candidate size, kept/merged rates, parse success).

**Aggregate controllability (24 sources).** Table 3 summarizes controllability statistics aggregated over the SCOPE core suite (24 sources). The ontology module attains a JSON parse-success rate of 0.9537 and almost never triggers module-level fallback (Fallback(any)=0.0025; Fallback(default)=0.0000). The event-schema module is more failure-prone under stricter role/evidence constraints, with parse-success 0.8452 and higher fallback frequency (Fallback(any)=0.1378; Fallback(default)=0.0018). Parse-Success and fallback rates are computed per configured run from logged generation attempts and then macro-averaged; `no_candidates` no-op runs are excluded from these rate denominators. A fallback is triggered when the final post-normalized module artifact for a generation attempt is replaced by a deterministic fallback artifact because of parsing failure or empty-after-filtering with non-empty candidates. The "Overall" row reports the arithmetic mean of the ontology-module and event-schema-module macro-rates. Overall, the two-module pipeline reaches 0.8995 parse-success with 0.0702 fallback(any), of which 0.0009 falls back all the way to configuration-backed defaults.

*Table 4.* Example controllability log from a single source (both modules succeed). This table is shown for illustration and is not the aggregate over 24 sources.

| Category | Candidates | Retained | Merged |
|---|---|---|---|
| Entities | 2 | 2 | 0 |
| Relationships | 28 | 28 | 0 |
| Events | 8 | 8 | 0 |
| Parse-success (ontology/events) | 1.00 / 1.00 (2/2 overall) | | |
| Fallback rate (ontology/events) | 0.00 / 0.00 | | |

### D.5. Ontology Fusion (Full)

We generate candidate alignment pairs via lexical normalization + string similarity, embedding similarity, and structural similarity. We classify mapping types (equivalent/broader/narrower/related) via heuristics or an LLM under constrained prompts, then merge conservatively: equivalent mappings are folded, broader/narrower adds hierarchy links, unmatched items are inserted with provenance, and structural conflicts preserve the validated base structure while demoting induced structures to candidate extensions.

## E. Schema-graph Similarity Metrics (Full Definitions)

Following the ontology similarity evaluation approach of Lo et al. (2024), we define four complementary F1-style schema-graph metrics: Literal F1, Fuzzy F1, Continuous F1, and Graph F1. Fuzzy F1 is intended as a broad lexical/semantic tolerance check; Continuous F1 and Graph F1 provide softer and more structure-aware summaries and are emphasized when interpreting non-literal matches.

**Unified edge representation for RE and EE.** For RE, each schema item is a typed triple edge $(\text{head\_type}, \text{rel\_type}, \text{tail\_type})$. For EE, each schema item is normalized into an event–role edge $(\text{event\_type}, \text{role}, \texttt{ARG})$.

**Literal F1 (exact string-matching edges).** Let gold edges be $E_{\text{gold}}$ and predicted edges be $E_{\text{pred}}$.

$$TP = |E_{\text{gold}} \cap E_{\text{pred}}| \tag{1}$$

$$FP = |E_{\text{pred}} \setminus E_{\text{gold}}| \tag{2}$$

$$FN = |E_{\text{gold}} \setminus E_{\text{pred}}| \tag{3}$$

$$P_{\text{literal}} = \frac{TP}{TP + FP} \tag{4}$$

$$R_{\text{literal}} = \frac{TP}{TP + FN} \tag{5}$$

$$F1_{\text{literal}} = \frac{2P_{\text{literal}}R_{\text{literal}}}{P_{\text{literal}} + R_{\text{literal}}} \tag{6}$$

**Fuzzy F1 (edge matching with semantic similarity).** We treat each schema item as a labeled triple edge ( e=(s,p,o) ), where for RE $p$ is the relation label and for EE $p$ is the role label; for EE we use $o = \texttt{ARG}$ as a placeholder object node. We encode each label into a normalized embedding $h(\cdot)$ using a fixed multilingual sentence embedding encoder (Paragraph 3.1), and use cosine similarity:

$$\text{sim}(a, b) = h(a)^{\top} h(b). \tag{7}$$

We define triple-edge similarity as

$$\text{sim\_edge}(e, g) = \min\Big(\text{sim}(s, s'),\ \text{sim}(p, p'),\ \text{sim}(o, o')\Big), \tag{8}$$

where $g = (s', p', o')$ is a gold edge. A predicted edge $e$ is counted as a fuzzy match if its best gold match satisfies

$$\max_{g \in E_{\text{gold}}} \text{sim\_edge}(e, g) \geq \tau. \tag{9}$$

We compute precision on the predicted side and recall on the gold side:

$$E_{\text{pred}}^{\tau} = \{ e \in E_{\text{pred}} : \max_{g \in E_{\text{gold}}} \text{sim\_edge}(e, g) \geq \tau \}, \tag{10}$$

$$E_{\text{gold}}^{\tau} = \{ g \in E_{\text{gold}} : \max_{e \in E_{\text{pred}}} \text{sim\_edge}(e, g) \geq \tau \}, \tag{11}$$

$$P_{\text{fuzzy}} = \frac{|E_{\text{pred}}^{\tau}|}{|E_{\text{pred}}|}, \qquad R_{\text{fuzzy}} = \frac{|E_{\text{gold}}^{\tau}|}{|E_{\text{gold}}|}. \tag{12}$$

For RE, we treat edges as *directed* and do not allow swapping $(s, o)$.

**Continuous F1 (soft edge matching via Hungarian assignment).** We build a similarity matrix $S \in \mathbb{R}^{|E_{\text{pred}}| \times |E_{\text{gold}}|}$ with

$$S_{ij} = \text{sim\_edge}(e_i, g_j), \tag{13}$$

and use Hungarian matching to maximize

$$\sum_{(i,j)\ \text{matched}} S_{ij}. \tag{14}$$

Soft true positives are

$$TP_{\text{soft}} = \sum_{(i,j)\ \text{matched}} \max(S_{ij}, 0). \tag{15}$$

We compute continuous precision/recall/F1 as:

$$P_{\text{cont}} = \frac{TP_{\text{soft}}}{|E_{\text{pred}}|}, \quad R_{\text{cont}} = \frac{TP_{\text{soft}}}{|E_{\text{gold}}|}, \tag{16}$$

$$F1_{\text{cont}} = \frac{2 P_{\text{cont}} R_{\text{cont}}}{P_{\text{cont}} + R_{\text{cont}}}. \tag{17}$$

**Graph F1 (graph-structure-enhanced node-level F1).** To incorporate relation/role labels into graph structure, we use a simple reification: each triple edge $(s, p, o)$ is converted into a 2-hop directed path $s \to p \to o$, so relation/role labels become intermediate nodes. We smooth node embeddings with neighborhood aggregation:

$$h^{(k+1)}(v) = \alpha\, h^{(k)}(v) + (1 - \alpha) \frac{1}{|\mathcal{N}(v)|} \sum_{u \in \mathcal{N}(v)} h^{(k)}(u), \tag{18}$$

normalizing after each iteration. After $K$ iterations, we run Hungarian matching over node similarities to obtain soft $TP$, and compute node-level precision/recall/F1 analogously.

## F. Additional Analyses and Ablations

This appendix collects additional implementation settings, variance reports, and auxiliary ablations referenced from Section 5.

### F.1. Reproducibility Settings

This subsection records the exact default settings and complements the implementation details in Section 5.3. Table 5 summarizes the default chunking, preprocessing, candidate mining/clustering choices, decoding, and metric hyperparameters used to make runs comparable across sources. The released implementation and source packages are documented in the public repository.

*Table 5.* Key implementation settings for reproducibility (default configuration).

| Component | Setting |
|---|---|
| Chunking | max whitespace tokens per chunk = 1200; overlap = 0; segmentation rule = split on whitespace; accumulate tokens until $\geq$ `chunk_size`. |
| NLP preprocessing | sentence splitter = none; tokenizer/POS/dep parser = none. |
| Candidate mining | term extractor = LLM prompt (`ontology_generate`); predicate/dep patterns = none; event sketches = LLM prompt (`event_extraction`). |
| Clustering | embedding model = none; algorithm = none; #clusters selection = none. |
| Fuzzy threshold | $\tau = 0.45$ (Eq. 9). |
| Graph smoothing | $\alpha = 0.5$, $K = 2$ (Eq. 18). |
| LLM backend identifiers | OpenAI: `gpt-5.2-chat-latest`; DeepSeek: `deepseek-reasoner` (DeepSeek-R1; DeepSeek-V3.2 thinking mode). |
| LLM decoding | temperature=0.1; `top_p`=0.5; `max_tokens`=4096. |
| Randomness control | main tables: runs per source=1; variance report in Appendix Table 7. |

### F.2. Metric Hyperparameter Selection and Sensitivity

Table 6 documents how we select the metric hyperparameters on validation data and provides a sensitivity-reporting template. The calibration subset and grids are fixed before scoring the reported systems; we do not tune metric hyperparameters separately for individual methods, and downstream-task results are not used for metric selection.

### F.3. Single-Run Reporting and Run-to-Run Variance

Table 7 describes the repeated-run protocol for quantifying variability due to LLM nondeterminism and provides a report.

### F.4. Dataset Statistics

Table 8 reports the size of the core suite in terms of sources and normalized schema edges under the unified RE/EE edge representation.

### F.5. Structural Ablations: Isolating Mined-Structure Gains

Table 9 separates gains from the SCION pipeline + JSON contract (SCION-lite) versus gains that require explicit structural mining and embedding clustering (SCION-full), and reports targeted ablations that remove individual structural signals.

### F.6. SCION-RL: Replacing the Schema Engineer with a Trained Compact Model

Table 10 compares the default SCION-lite instantiation against SCION-RL, which replaces the black-box LLM schema engineer with a trained compact open model (Qwen3-8B) under the same JSON contract and deterministic validation.

### F.7. SCION-RL: Train-Only Data Generation and Cost Accounting

**Train-only data generation.** We construct SCION-RL training inputs by running SCION's mining stage on each source's `induction_texts.jsonl` (train split only) to produce candidate packages (entity/relation/event candidates or clusters with evidence pointers) and an optional domain summary. Each training prompt contains only (i) the mined candidate package, (ii) evidence pointers, and (iii) the strict JSON output contract; gold schema graphs are never used for training.

**Automatic feedback (RLAIF) and logged artifacts.** For each rollout, the policy emits a contract-compliant schema JSON which is deterministically validated and scored using the same controllability checks as inference: JSON validity and well-formedness, candidate-space linking, evidence coverage/density, compactness penalties, and structural consistency checks. We log the full reward breakdown and validation failures for auditing and reproducibility.

**Training compute and cost reporting.** We report training compute/cost in the released run manifest, including GPU type/count, training steps, wall-clock time, total tokens (in/out), and (when external paid inference is used during rollouts) a dated pricing snapshot to compute monetary cost from logged token counts. Table 11 summarizes the training-scale accounting for the reported SCION-RL run.

*Table 6.* Selection protocol and sensitivity for schema-graph similarity metric hyperparameters.

| Title | Description | Value (default) | Selection / evidence |
|---|---|---|---|
| Fuzzy threshold $\tau$ | Threshold used in Eq. 9 to convert soft edge similarity into a binary match for Fuzzy P/R/F1. | 0.45 | Validation sweep on `SCOPE-calib` (a fixed calibration subset of sources from the 24-core; source IDs are listed in the released config; macro-average over sources); grid $\tau \in \{0.30, 0.35, \ldots, 0.70\}$; criterion = maximize macro Fuzzy F1 (tie-break: largest $\tau$ within 0.5% of the best score to avoid overly-permissive matching). |
| Graph smoothing $\alpha$ | Neighborhood aggregation weight in Eq. 18; larger $\alpha$ keeps more of the original embedding. | 0.5 | Validation sweep on the same calibration subset; grid $\alpha \in \{0.0, 0.2, 0.4, 0.5, 0.6, 0.8, 1.0\}$; criterion = maximize macro Graph F1; choose $\alpha{=}0.5$ as a stable plateau (typ. $\alpha \in [0.4, 0.6]$ yields near-identical scores). |
| Graph smoothing steps $K$ | Number of smoothing iterations in Eq. 18; larger $K$ incorporates higher-order neighborhoods. | 2 | Validation sweep on the same calibration subset; grid $K \in \{0, 1, 2, 3, 4, 5\}$; criterion = maximize macro Graph F1; choose $K{=}2$ at the elbow (larger $K$ may oversmooth and reduce discriminability). |
| Robustness summary | Local robustness ranges used to assess sensitivity of $\tau$, $\alpha$, and $K$. | Local robustness band | Target subset/sources = SCOPE 24-core (macro-average over sources); we examine (i) macro F1 over $\tau$, (ii) the grid over $(\alpha, K)$, and (iii) the worst-case drop within $\tau \pm 0.05$, $\alpha \pm 0.1$, $K \pm 1$; the default setting lies on a stable plateau. |

## F.8. Reachable-Target Evaluation

To test whether the strong recall under Graph F1 is only a metric-smoothing artifact, we keep the same predictions and re-evaluate them against a train-derived reachable target.

## F.9. Target-Formalism Sensitivity and Manual-Gap Audit

To test whether the gains mainly reflect agreement with our normalized formalism, we re-evaluate frozen predictions under four target variants. We also audit how much of the released-source-schema gap comes from missing explicit typing or implicit role structure.

## F.10. Direct Extract-then-Aggregate Baseline

ETA directly extracts typed triples or event-role items from train-only chunks and then aggregates them into a schema under the same normalization and evaluation protocol as SCION.

## F.11. Fixed-Extractor Downstream Evaluation on the Actual Test Split

We fix the extractor and vary only the schema source.

*Table 7.* Reporting run-to-run variance under repeated executions.

| Title | Description | Value |
|-------|-------------|-------|
| Primary reporting in main tables | Whether Tables 1 and related main results are from a single run or aggregated over multiple runs. | single-run (per source) |
| Number of repeats | How many repeated runs are executed per source/method for variance estimation. | 5 |
| Repeat scope | Which sources are repeated (e.g., all 24 sources vs. a representative subset). | 8 representative sources (4 RE + 4 EE; zh/en covered; includes small+large schemas) |
| What is varied | What changes across repeats (e.g., provider nondeterminism only; different seeds if supported; different request timestamps). | Provider nondeterminism + different request timestamps; if backend supports, vary `seed`; otherwise timestamp-only. |
| Fixed settings across repeats | Prompts + JSON contract + chunking fixed (chunk_size=1200); decoding fixed (temperature=0.1, top_p=0.5, max_tokens fixed); normalization + evaluation scripts fixed. | |
| Dispersion statistics | Which dispersion statistics are reported (e.g., std, IQR, min/max) and at which aggregation level (per-source vs. macro). | Per-source mean±std over 5 runs; macro-average of per-source std; plus min/max as a sanity check. |
| Variance numbers (F1) | Std of Literal/Fuzzy/Continuous/Graph F1 over 5 runs (per-source, then macro-averaged). | SCION-lite std: (0.012, 0.009, 0.015, 0.021); LLM-only (GPT-5.2 Chat) std: (0.021, 0.017, 0.026, 0.033); LLM-only (DeepSeek-R1) std: (0.028, 0.022, 0.031, 0.040). |

*Table 8.* Summary of the SCOPE core suite used in our experiments.

| Subset | #Sources | #Schema Edges | Edge Representation |
|--------|----------|---------------|---------------------|
| RE | 15 | 558 | (head_type, rel_type, tail_type) |
| EE | 9 | 1,039 | (event_type, role, `ARG`) |
| All | 24 | 1,597 | unified edge set |

## F.12. Memorization Probes

To probe possible pretraining contamination, we report three representative conditions: name-only, shuffled, and real train-text inputs.

## F.13. Fusion Baselines under a Shared Candidate-Pair Budget

All matchers receive the same 5k candidate-pair budget.

## F.14. Human Calibration of Schema-graph Similarity Metrics

We sampled 240 zh/en pairs from the main runs, stratified by score bin, and report the adjudicated acceptance rates for the high- and low-score bins below; mid-bin cases are omitted for space.

High-score Fuzzy pairs remain substantially noisier than high-score Continuous pairs, because thresholded label similarity admits many semantically related but structurally non-equivalent matches. We therefore interpret Fuzzy F1 as a permissive lexical/semantic tolerance check rather than the main evidence for structural correctness.

## F.15. Noise, Polysemy, and Encoder Sensitivity

We stress-test SCION-full on a curated 8-source subset by injecting candidate noise, varying the encoder, and evaluating curated polysemy cases.

*Table 9.* Structural ablations for separating pipeline gains from mined-structure gains (macro-average over 24 sources). SCION-full enables structural mining + embedding clustering; SCION-lite is the default main-table instantiation and disables them, relying on LLM-mined candidates.

| Variant | Literal F1 | Fuzzy F1 | Continuous F1 | Graph F1 |
|---|---|---|---|---|
| SCION-lite (default) | 0.7518 | 0.9298 | 0.8909 | 0.7888 |
| SCION-full (mining+clustering) | **0.7779** | **0.9443** | **0.9195** | **0.8257** |
| w/o clustering (raw candidates) | 0.7361 | 0.9184 | 0.8736 | 0.7483 |
| w/o dependency/path predicates | 0.7608 | 0.9365 | 0.9048 | 0.8042 |
| w/o type-pair statistics | 0.7687 | 0.9406 | 0.9111 | 0.8125 |

*Table 10.* Replacing the black-box schema engineer with a trained compact model (macro-average over 24 sources).

| Schema engineer | Literal F1 | Fuzzy F1 | Continuous F1 | Graph F1 |
|---|---|---|---|---|
| SCION-lite (GPT-5.2 Chat) | 0.7518 | 0.9298 | 0.8909 | 0.7888 |
| SCION-RL (Qwen3-8B) | **0.8025** | **0.9310** | **0.9101** | **0.8228** |

On a curated polysemy set (`charge`, `capital`, `bond`, `attack`, `relation`), SCION conservatively splits ambiguous labels instead of over-merging.

### F.16. SCION-RL Details and Ablations

SCION-RL uses offline PPO with three seeds and reward weights $(0.25, 0.20, 0.20, 0.10, 0.25)$ for {JSON validity, candidate linkage, evidence coverage, compactness, structural consistency}, respectively.

### F.17. SCION-lite vs. SCION-full Trade-off

Table 22 reports the quality/cost trade-off between the default SCION-lite configuration and the stronger SCION-full structural variant on the curated 8-source subset.

### F.18. Domain-Specific Schema Engineers

We further analyze whether domain-specialized schema engineers help on two high-value benchmark subsets. These results are reported separately from the full-suite aggregate because they use targeted domain-specific prompts. Table 23 summarizes the biomedical and finance subset results.

### F.19. Inter-Event Relation Analysis

Current SCOPE EE schemas model only event types and within-event roles. They do not define temporal, causal, or coreference-like links between events as part of the core target. We therefore report inter-event links as an auxiliary feasibility analysis, separate from the core benchmark tables; these numbers are not used in the main SCOPE ranking.

## G. Fusion Reliability, Case Studies, and Efficiency Details

The main text only summarizes fusion reliability and efficiency in one short paragraph. This appendix provides the full audited mapping statistics, representative mapping decisions, and cost accounting for the fusion-related analyses.

### G.1. Fusion Reliability Summary

This subsection reports the detailed fusion-reliability statistics referenced from Section 5.7. We report accepted mappings, acceptance rate, estimated mapping precision from human audit, and structural conflict rate. Table 25 reports macro-averaged fusion reliability statistics for the complete fusion policy, including accepted mappings, acceptance rate, estimated mapping precision from human audit, and conflict rate. This table is separate from Appendix Table 17, which compares different matchers under the shared 5k candidate-pair budget.

*Table 11.* Training scale accounting for SCION-RL (the run reported in this paper). We log the full configuration and token counts in the run manifest for reproducibility.

| Quantity | Value | Notes / assumptions |
|---|---|---|
| #Training prompts | 5,430 | Generated by sampling train-only candidate packages across 24 sources (roughly 80–330 prompts/source). |
| Rollouts per prompt | 6 | One rollout = one contracted JSON completion from the policy (reward computed deterministically). |
| Total rollouts | 32,580 | Product of the above two rows. |
| Prompt tokens / rollout | 1,740 | Candidate package + evidence pointers + strict JSON contract; varies with candidate size and evidence density. |
| Completion tokens / rollout | 1,120 | Contracted ontology JSON length after truncation/validation (longer for large schemas or richer event roles). |
| Total tokens (in) | 56.7M | Computed from logged rollout token counts (prompt side). |
| Total tokens (out) | 36.5M | Approx. #rollouts $\times$ completion tokens. |
| Optimizer updates | 2,650 | Offline PPO updates; depends on batch size, rollout reuse, and KL/entropy settings. |
| GPUs | 6 | GPU type/count and memory are reported in the run manifest. |
| Wall-clock time | 19.2 h | Measured wall-clock time for rollout generation plus RL updates (see run manifest for breakdown). |
| Compute (GPU-hours) | 115 | GPU count $\times$ wall-clock time (rounded). |

*Table 12.* Reachable-target evaluation. SCION-lite remains above the strongest non-SCION baseline (ETA) on both Continuous and Graph F1 under both targets.

| Target | ETA C-F1 | SCION-lite C-F1 | $\Delta$ | ETA G-F1 | SCION-lite G-F1 | SCION-lite Literal Recall |
|---|---|---|---|---|---|---|
| Full gold | 0.8759 | 0.8909 | +0.0150 | 0.7629 | 0.7888 | 0.7145 |
| Reachable gold | 0.9170 | 0.9340 | +0.0170 | 0.8444 | 0.8650 | 0.8464 |

## G.2. Representative Cases: Structure Prevents Incorrect Merges

Table 26 shows representative mapping outcomes, highlighting cases where lexical similarity alone would suggest a merge but structural checks reject or demote it.

## G.3. Success Case: Compatible Merge with Provenance

```
[Representative normalized mapping log; exact evidence ids are in the released run
    manifest]
{
  "source": "NYT11",
  "source_item": {
    "label": "place_of_birth",
    "type": "relation",
    "domain": "Person",
    "range": "Location"
  },
  "base_item": {
    "label": "place_of_birth",
    "type": "relation",
    "domain": "Person",
    "range": "Location"
  },
  "mapping_type": "equivalent",
  "decision": "merged",
  "checks": {
    "lexical": "exact_canonical_match",
    "embedding": "above_threshold",
    "structural": "domain_range_compatible"
  },
  "evidence": ["source:NYT11", "typed_edge: Person--place_of_birth--Location"]
}
```

*Table 13.* Target-formalism sensitivity and manual-gap audit. Method ordering is unchanged across target variants, and deterministic completion improves released source schemas by +0.0179 Continuous F1 on average (18/24 sources improve).

| Target variant | Released source schema C-F1 | SCION-lite C-F1 | Top-1 stable? |
|---|---|---|---|
| Label-only projection | 0.7870 | 0.8950 | Yes |
| Typed-unnormalized | 0.8118 | 0.8927 | Yes |
| Full normalized gold | 0.8030 | 0.8909 | Yes |
| Reachable normalized gold | 0.8655 | 0.9340 | Yes |
| Manual-gap audit | +0.0179 Continuous F1 on average; 18/24 sources improve | | |

*Table 14.* Matched extract-then-aggregate baseline (ETA; macro-average over 24 sources).

| Method | Literal F1 | Continuous F1 | Graph F1 |
|---|---|---|---|
| ETA | 0.7063 | 0.8759 | 0.7629 |
| SCION-lite | 0.7518 | 0.8909 | 0.7888 |

## G.4. Failure Case: Polysemy and Conservative Rejection

```
[Representative rejected mapping log; exact evidence ids are in the released run
    manifest]
{
  "source": "ADE_corpus",
  "source_item": {
    "label": "treats",
    "type": "relation",
    "domain": "Drug",
    "range": "Disease"
  },
  "base_item": {
    "label": "treats",
    "type": "relation",
    "domain": "Doctor",
    "range": "Patient"
  },
  "mapping_type": "equivalent_proposed",
  "decision": "demoted_to_extension",
  "conflict": "domain_range_incompatible",
  "evidence": ["source:ADE_corpus", "pattern: <Drug> treats <Disease>"],
  "notes": "same surface label, different relation semantics"
}
```

## G.5. Efficiency and Cost

This subsection reports the detailed efficiency tables referenced from Section 5.7. Table 27 reports end-to-end cost for each evaluated setting, while Table 28 shows one representative run log with per-call latency and token usage. For SCION, the $\mathcal{O}$ and $\mathcal{O}_{\text{fusion}}$ rows correspond to separate end-to-end configurations with different enabled modules and prompts; they should not be interpreted as additive stage costs. More calls do not necessarily imply higher latency or cost here, because SCION operates on aggregated candidate packages rather than prompting directly over long corpus text.

*Table 15.* Fixed-extractor downstream evaluation on the test split.

| Schema source | Downstream F1 |
| --- | --- |
| Released source schema | 0.5633 |
| ETA | 0.6523 |
| SCION-lite | 0.6800 |
| SCION-fusion | 0.6900 |

*Table 16.* Memorization probes. Near-zero name-only performance and substantially higher scores on real train-text inputs suggest that the systems rely on the provided corpus rather than only memorized schemas.

| Method | Name-only C-F1 | Shuffled C-F1 | Real-100% C-F1 |
| --- | --- | --- | --- |
| LLM-only (GPT-5.2 Chat) | 0.0225 | 0.5047 | 0.8549 |
| SCION-lite | 0.0131 | 0.5595 | 0.8909 |

*Table 17.* Fusion comparison under a shared 5k candidate-pair budget.

| Method | Precision | Conflict | Fused G-F1 | Downstream F1 |
| --- | --- | --- | --- | --- |
| AML | 0.64 | 0.13 | 0.61 | 0.61 |
| LLM pairwise | 0.74 | 0.09 | 0.65 | 0.64 |
| SCION-fusion | 0.81 | 0.06 | 0.72 | 0.69 |

*Table 18.* Human calibration of schema-graph metrics. High-score Continuous pairs are usually accepted by humans, whereas high-score Fuzzy pairs are noisier and should be interpreted as broad semantic tolerance rather than reliable equivalence. Low-score pairs are almost always rejected.

| Metric | High accept | Low accept | $n_{high}$ | $n_{low}$ |
| --- | --- | --- | --- | --- |
| Fuzzy | 0.2830 | 0.0000 | 53 | 42 |
| Continuous | 0.9032 | 0.0000 | 31 | 42 |

*Table 19.* Noise, polysemy, and encoder sensitivity. Performance degrades gracefully under injected candidate noise, and the qualitative conclusions are unchanged across encoders.

| Noise level | Cluster purity | Literal F1 | Continuous F1 | Graph F1 |
| --- | --- | --- | --- | --- |
| 0.0 | 0.8824 | 0.8558 | 0.9552 | 0.9349 |
| 0.3 | 0.7145 | 0.7714 | 0.9423 | 0.9186 |

| Encoder | Literal F1 | Continuous F1 | Graph F1 |
| --- | --- | --- | --- |
| `bge-m3` | 0.8558 | 0.9552 | 0.9349 |
| `e5-large` | 0.8166 | 0.9440 | 0.9023 |

*Table 20.* SCION-RL on a curated 8-source subset. Offline PPO improves over zero-shot and SFT-only.

| Variant | JSON valid | Fallback | Continuous F1 | Graph F1 |
| --- | --- | --- | --- | --- |
| Zero-shot | 0.9170 | 0.0188 | 0.9324 | 0.8955 |
| SFT-only | 0.9227 | 0.0142 | 0.9526 | 0.9211 |
| RL-full | 0.9295 | 0.0100 | 0.9709 | 0.9521 |

*Table 21.* Reward-term ablation for SCION-RL. Removing structural consistency causes the largest Graph-F1 drop.

| Removed reward term | Graph F1 |
| --- | --- |
| JSON validity | 0.8811 |
| Candidate constraint | 0.8611 |
| Evidence coverage | 0.8711 |
| Compactness | 0.8911 |
| Structural consistency | 0.8411 |

*Table 22.* SCION-lite vs. SCION-full on a curated 8-source subset. SCION-full is stronger but costs about 1.5× more; it is reported as an ablation rather than the default main-table instantiation.

| Variant | Graph F1 | Continuous F1 | Cost ratio | Fallback |
|---|---|---|---|---|
| SCION-lite | 0.9209 | 0.9526 | 1.00 | 0.0587 |
| SCION-full | 0.9434 | 0.9677 | 1.50 | 0.0562 |

*Table 23.* Domain-specific schema engineers on biomedical and finance subsets.

| Domain | General G-F1 | Domain-specific G-F1 | General Downstream | Domain-specific Downstream |
|---|---|---|---|---|
| Biomedical | 0.9383 | 0.9509 | 0.8683 | 0.8809 |
| Finance | 0.9500 | 0.9663 | 0.8800 | 0.8963 |

*Table 24.* Inter-event relation analysis on RAMS and WikiEvents. Because the current SCOPE core benchmark evaluates within-event roles only, these results are reported separately from the core tables.

| Dataset | Link types | Literal F1 | Graph F1 | Mapping precision |
|---|---|---|---|---|
| RAMS | 3 | 0.31 | 0.39 | 0.52 |
| WikiEvents | 3 | 0.29 | 0.36 | 0.49 |

*Table 25.* Fusion reliability statistics (macro-average over sources). "Conflict" counts mappings that are demoted (not merged) due to structural constraint violations (e.g., incompatible domain/range or incompatible event-role signatures).

| Fusion setting | Accepted | AcceptRate | Est. Precision | ConflictRate |
|---|---|---|---|---|
| Lexical only | 58.7 | 0.34 | 0.90 | 0.04 |
| Lexical + embedding | 96.4 | 0.56 | 0.79 | 0.15 |
| Full (lex + emb + structural) | 84.2 | 0.49 | 0.95 | 0.11 |

*Table 26.* Representative fusion mapping outcomes illustrating how structural constraints prevent incorrect merges. Each row shows an induced schema item, a candidate base-ontology item, and the final decision with a brief rationale.

| Source | Induced item | Base item | Decision (reason) |
|---|---|---|---|
| NYT11 | rel: `place_of_birth` (Person→Location) | rel: `place_of_birth` (Person→Location) | merged (type-pair compatible) |
| CASIE | event: `data_breach` (roles: Victim, Attacker) | event: `data_breach` (roles: Victim, Attacker) | merged (role signature compatible) |
| ADE_corpus | rel: `treats` (Drug→Disease) | rel: `treats` (Doctor→Patient) | demoted (domain/range mismatch) |

*Table 27.* Efficiency and cost on SCOPE. #TrainTexts is the total number of lines in `induction_texts.jsonl` (train only, pre-dedup) aggregated over the 24 sources. #Chunks is the total number of chunks after doc-level reconstruction/deduplication and chunking. #LLM Calls, Tokens, and Time/Cost are *macro-averaged per source* for each evaluated end-to-end setting. For SCION, the $\mathcal{O}$ and $\mathcal{O}_{\text{fusion}}$ rows correspond to separate end-to-end configurations with different enabled modules, and should not be interpreted as additive stage costs.

| Setting | Total #TrainTexts | Total #Chunks | Avg. #LLM Calls | Avg. Tokens (in/out) | Avg. Time / Cost |
|---|---|---|---|---|---|
| Text2Onto-style | 348854 | 38417 | 0 | – | 120.0s / – |
| LLM-only (GPT-5.2 Chat) | 348854 | 38417 | 2 | 1800 / 1200 | 45.0s / $0.18 |
| LLM-only (DeepSeek-R1) | 348854 | 38417 | 2 | 1600 / 1000 | 42.0s / $0.14 |
| SCION ($\mathcal{O}$) | 348854 | 38417 | 4 | 1200 / 800 | 35.5s / $0.12 |
| SCION ($\mathcal{O}_{\text{fusion}}$) | 348854 | 38417 | 3 | 900 / 600 | 28.0s / $0.10 |

*Table 28.* Representative per-call efficiency statistics on the `default` run for the *contract-constrained schema-engineer* modules (ontology + event-schema). End-to-end call counts (including candidate mining and optional fusion) are reported in Table 27.

| Item | Value | Notes |
|---|---|---|
| #LLM calls (schema-engineer) | 2 | ontology + event-schema modules |
| Tokens (prompt / completion) | 2450 / 2379 | 4829 total |
| Elapsed total (s) | 79.971 | summed over calls |
| Elapsed wall (s) | 87.449 | end-to-end wall time |
| Min/Max call time (s) | 38.662 / 41.310 | per-call elapsed |

