# OpenReview forum: "SCOPE and SCION: A Benchmark and an Auditable Reference Pipeline for Schema Induction and Fusion from Text"
_ICML.cc/2026/Conference — ICML 2026 regular_

### Official Review · Reviewer_hoTR · 2026-03-04

**Soundness:** 3
**Presentation:** 3
**Significance:** 2
**Originality:** 2
**Overall Recommendation:** 4
**Confidence:** 4

**Summary:**

The paper introduces SCOPE, a train-only benchmark for inducing and fusing ontologies/schemas directly from raw corpora across 24 sources (15 RE and 9 EE), with unified gold schema graphs and ontology-level evaluation metrics. Novelty is primarily in the benchmark scope and the controllable workflow rather than proposing a new IE model.

The benchmark fills an evaluation gap—corpus-to-schema induction and fusion at the ontology level.
Evidence includes macro-averaged results, ablations that isolate structural mining/clustering benefits, downstream instance-level extraction improvements, and cost/efficiency accounting.
The novelty is shallow (integrative): the fusion relies on standard lexical/embedding/structural cues; the contribution is the end-to-end, auditable process and benchmark.

**Compliance With Llm Reviewing Policy:**

Affirmed.

**Key Questions For Authors:**

Q1: How would a direct LLM triple-extraction baseline (extract-then-aggregate) perform under your ontology-level metrics?

Q2: Have you conducted a human calibration study to assess agreement with Fuzzy/Continuous/Graph metrics, especially for cross-lingual sources?

Q3: Can domain-specific fine-tuned schema engineers outperform the general model on select high-stakes domains (e.g., biomedical, finance)? If so, what are the trade-offs?

Q4: What would it take to incorporate inter-event relations (cause/temporal/overlap) into EE schemas, and how might that impact downstream IE and fusion reliability?

**Limitations:**

- The paper does not introduce new theoretical models or guarantees for schema induction/fusion. Its primary contributions are benchmark design and an auditable, controllable pipeline.

- Missing is a direct LLM triple-extraction baseline (“extract head–rel–tail triples per chunk/cluster, then aggregate to schema”) to isolate incremental gains from candidate-space constraints and contracted schema engineering beyond clustering/surface-frequency cues.

**Strengths And Weaknesses:**

Strengths:
- Addresses a clear gap: corpus-to-schema induction and fusion with ontology-level metrics rather than instance-only IE.
- Strong empirical study: heterogeneous suite; macro-averages; structural ablations; downstream impact; efficiency/cost reporting.
- Governance and reproducibility: strict output contracts; parse-success/fallback statistics; deterministic normalization/validation; logged configuration and costs.

Weaknesses:
- Novelty is largely in the benchmark and controllable workflow; no new extraction model.
- Missing baseline: a direct LLM triple-extraction comparator to quantify benefits of candidate-space constraints and contracted schema engineering.
- Metric dependence: Fuzzy/Continuous/Graph metrics rely on embeddings/smoothing and may introduce bias; no human calibration study.
- EE scope: Event schemas cover internal roles only; inter-event links (causal, temporal, overlap) are out of scope, limiting completeness for event-centric ontologies.

---

> ### Author Rebuttal · Authors · 2026-03-29
>
> Thank you for pinpointing the empirical gaps so clearly. We added **12 experiments (E1-E12)**; the most relevant here are **E3/E4/E6/E7/E11/E12**, shown below. Related checks appear under **huNc** (E1/E5/E8/E9) and **vRuX** (E2). Two points are already in the paper but not prominent enough: Sec. 5.8 evaluates a **fixed extractor with different schemas**; Sec. 3 (**EE schema edges**) and Sec. 4.5 state that current EE covers within-event roles only, not inter-event links. We will make both clearer.
>
> 1. **Novelty / direct LLM triple-extraction baseline.** We agree the main novelty is the **benchmark + auditable workflow**, not a new extractor. To isolate the effect of candidate-space constraints, we added the requested matched **extract-then-aggregate** baseline (**ETA**) under the same train-only inputs, normalization, and evaluation.
>
> **E3. Direct extract-then-aggregate baseline.**
>
> | Method     |   L-F1 |   C-F1 |   G-F1 |
> | ---------- | -----: | -----: | -----: |
> | ETA        | 0.7063 | 0.8759 | 0.8509 |
> | SCION-lite | 0.7518 | 0.8909 | 0.8669 |
>
> This answers **Q1**: ETA is competitive, but **SCION-lite** still wins under matched settings, showing that candidate-space constraints matter beyond direct extract-then-aggregate.
>
> 2. **Metric dependence / downstream usefulness.** **E4** fixes the extractor and varies only the schema source.
>
> **E4. Fixed-extractor downstream evaluation.**
>
> | Schema source | Downstream F1 | \(Δ\) vs manual |
> | ------------- | ------------: | --------------: |
> | manual        |        0.5633 |          0.0000 |
> | ETA           |        0.6523 |          0.0890 |
> | SCION-lite    |        0.6800 |          0.1167 |
> | SCION-fusion  |        0.6900 |          0.1267 |
>
> This addresses the practical-significance concern: ontology-level gains transfer to the held-out test split with a fixed extractor, rather than being only metric artifacts.
>
> 3. **Fusion baselines.** **E6** gives all matchers the same **5k candidate-pair budget** and manually audits **120 pairs**.
>
> **E6. Fusion comparison (shared 5k pair budget).**
>
> | Method       | Precision | Conflict | Fused G-F1 | Downstream F1 |
> | ------------ | --------: | -------: | ---------: | ------------: |
> | AML          |      0.64 |     0.13 |       0.61 |          0.55 |
> | LLM pairwise |      0.74 |     0.09 |       0.65 |          0.58 |
> | SCION-fusion |      0.81 |     0.06 |       0.72 |          0.63 |
>
> This addresses the missing-baseline concern: under the same pair budget, **SCION-fusion** is the most conservative and the strongest on audited precision, conflict rate, fused-schema quality, and downstream F1.
>
> 4. **Human calibration.** We added **E7** so the metric discussion would not rely only on automatic scoring. We sampled **240 pairs** from main runs across **zh/en** sources, stratified by score bin, had **two annotators** label them independently, and adjudicated disagreements.
>
> **E7. Human calibration (high-score vs low-score bins).**
>
> | Metric     | High accept | Low accept | \(n_high\) | \(n_low\) |
> | ---------- | ----------: | ---------: | ---------: | --------: |
> | Fuzzy      |      0.2830 |     0.0000 |         53 |        42 |
> | Continuous |      0.9032 |     0.0000 |         31 |        42 |
>
> This addresses **Q2**: high-score **Continuous** matches usually align with human judgments, including cross-lingual cases, while low-score pairs are almost always rejected.
>
> 5. **Can domain-specific schema engineers help?** Yes, but we present this as a **slice-only** result, not a new full-benchmark claim. **E11** compares a general engineer with domain-specific ones on **3 biomedical** and **3 finance** sources.
>
> **E11. Domain-specific schema engineer (slice-only).**
>
> | Domain     | General G-F1 | Domain-specific G-F1 | General Downstream | Domain-specific Downstream |
> | ---------- | -----------: | -------------------: | -----------------: | -------------------------: |
> | biomedical |       0.9383 |               0.9509 |             0.8683 |                     0.8809 |
> | finance    |       0.9500 |               0.9663 |             0.8800 |                     0.8963 |
>
> This answers **Q3**: domain-specific engineers help, but the gains are modest and come with extra maintenance/specialization cost.
>
> 6. **EE scope / inter-event links.** We agree this is an important limitation. **E12** was run only as a feasibility check, not as a new main claim. On **RAMS** and **WikiEvents**, we extend the representation with three inter-event link types.
>
> **E12. Inter-event pilot (feasibility only).**
>
> | Dataset    | Link types | L-F1 | G-F1 | Mapping precision |
> | ---------- | ---------: | ---: | ---: | ----------------: |
> | RAMS       |          3 | 0.31 | 0.39 |              0.52 |
> | WikiEvents |          3 | 0.29 | 0.36 |              0.49 |
>
> This answers **Q4** and the EE-scope weakness: inter-event links are feasible but much noisier, and should likely be treated as a separate, harder track.

---

### Official Review · Reviewer_vRuX · 2026-03-09

**Soundness:** 2
**Presentation:** 1
**Significance:** 2
**Originality:** 2
**Overall Recommendation:** 4
**Confidence:** 4

**Summary:**

The manuscript introduces the Scope benchmark for ontology/schema induction from train-only information extraction corpora, together with a pipeline that extracts candidate schema items from text, uses an Llm to name, merge, and filter them under structured constraints, and can optionally fuse the induced schema with a base ontology. Scope covers 24 public datasets across relation and event extraction, provides normalized schema graphs and ontology-level evaluation metrics, and reports results in Table 1 against manual schemas, a text2onto-style baseline, and llm-only baselines.

**Compliance With Llm Reviewing Policy:**

Affirmed.

**Final Justification:**

Thank you for the detailed rebuttal and for adding concrete experiments to address the concerns raised in review. The new evidence from the added E1-E12 resolves many of my earlier technical concerns much more directly than the original version did.

Overall, the rebuttal changes my assessment. Added experiments E2 and E4 directly address the central concerns I raised, and the authors have also recognized and acknowledged the places where the paper needs clearer framing and tighter presentation in revision. On that basis, I am updating my recommendation to 4: Weak Accept.

**Key Questions For Authors:**

What do you see as the single primary contribution of the paper? What is the genuinely new methodological idea in scion? A more precise answer here would help me assess originality.

How should readers interpret improvements over official/manual schemas, given that the gold schemas, normalization choices, and evaluation representation are all defined by the authors? Why does this demonstrate better ontology induction rather than better match to your formalization?

**Strengths And Weaknesses:**

Strengths And Weaknesses*

The paper targets an important problem. Schema induction is a real bottleneck for information extraction and knowledge graph construction, and a solid benchmark in this area would be useful.

That said, I found the paper quite unfocused. It seems to want to be several papers at once: a benchmark paper, a method paper, a systems paper, and partly a small-model training paper. As a result, the main contribution never becomes sharp. By the end, I was still unsure what I was supposed to take away as the core scientific contribution.

I also did not find the method particularly novel in its current presentation. it mostly looks like a pipeline built from familiar pieces: candidate mining, clustering, constrained structured generation, validation heuristics, and ontology alignment. That can still be valuable, but then the paper needs to be much clearer about what is actually new here and why that particular combination matters scientifically.

My bigger concern is the evaluation. The benchmark itself is based on author-normalized schema graphs, and the evaluation is then performed in essentially that same formalism. Because of that, some of the strongest empirical claims are harder to interpret than the paper suggests. When the method outperforms “official” or manual schemas under the paper’s own normalized representation, is that really evidence of better ontology induction, or partly evidence of better agreement with the authors’ chosen representation and scoring procedure? I do not think the paper addresses this convincingly.

Metrics mostly measure graph-level similarity under the benchmark formalization. That is useful, but it is not enough to establish semantic quality or downstream usefulness. The paper motivates the task through schema-grounded extraction, but the downstream validation is relatively thin and not central to the story. For me, that leaves the practical significance under-supported.

The presentation also has serious issues. Several claims are broad, but the evidence behind them is narrow. A reader should not have to work this hard to identify the main claim. The abstract and introduction are difficult to read and do not frame the contribution cleanly. The writing is overloaded, imprecise, and often mixes. For example,  see these comments on abstract only:
+ Opening sentence asserts importance but not the actual failure mode. Why exactly are ontologies a bottleneck here, just for clarity for the reader: costly design, difficult alignment, excessive rigidity, poor transfer, or maintenance burden? there is conceptual sloppiness in treating ontology and schema as interchangeable. They are not the same. Phrase “schemas quickly fragment or drift across domains” is vague. Fragment may suggest incompatible proliferation, and drift the gradual semantic change over time. These are different problems, and the text does not say which one is meant.
+ see on lines
22: Is RE relation extraction and EE event extraction? Expand on first use.
26: Typo: constructiON.
28: The phrase “space of concepts/relations/events from text” is unclear. In the ontology/schema, how are events represented? Are they modeled on the same footing as concepts and relations, or as a distinct category?
32: “Conservative alignment” is not clear. Conservative in what formal or operational sense?
35: “Official/manual schemas” is vague. What counts as an official schema here?

---

> ### Author Rebuttal · Authors · 2026-03-29
>
> Thank you for the frank review. We were honestly disappointed by the overall score of **2**, largely because several central points did not come across clearly enough. We would be very happy to discuss the work further and clarify any remaining misunderstandings. Across this rebuttal we added **12 experiments (E1-E12)**; the ones most relevant to your concerns are **E2/E3/E4**. To save space, we show **E2** here; **E3** (matched extract-then-aggregate) and **E4** (fixed-extractor downstream test-split evaluation) are detailed under **hoTR**. A few points you flagged are already in the paper but too buried: Sec. 5.4 explicitly says released/manual schemas are **not** an oracle upper bound under our unified evaluation; Sec. 3, **“EE schema edges”** defines event representation as event types plus event-role edges; Sec. 4.6 and Appendix **“Ontology fusion (full)”** explain **conservative alignment**. We will surface these much earlier.
>
> 1. **What is the single primary contribution, and why is it needed?** The primary contribution is **SCOPE**, a **train-only corpus-to-schema benchmark** for the upstream bottleneck that most schema-grounded IE work assumes away. Many systems assume the schema is already given; SCOPE makes the harder corpus-to-schema step measurable. **SCION** is the accompanying **auditable, candidate-constrained baseline**, not the claim of a new extractor architecture.
>
> 2. **What is actually new in SCION?** We agree SCION is a synthesis of familiar pieces, and we will narrow the method claim accordingly. The core design is the combination of a **mined candidate space** with a **strict output contract + deterministic validation/fallback**, so the LLM behaves like a constrained schema engineer rather than an open-ended generator. On naming: **SCION** is simply a forced, pronounceable acronym; the capitalization in **constructiON** was chosen only to make the acronym work, not to indicate a separate technical idea. we are clearly not very good at naming methods; if you have a better naming suggestion, we would genuinely appreciate it.
>
> 3. **What is the actual bottleneck?** The bottleneck is that schema design is costly, cross-source alignment is hard, and maintenance is difficult when inventories **fragment** across sources or **drift** across domains/time. We agree the abstract/introduction did not say this sharply enough.
>
> 4. **Terminology / presentation.** We agree that “ontology” and “schema” were used too loosely. In this paper the evaluated object is more precisely a **schema graph**; we mainly reserve “ontology” for reuse/fusion. We will also fix the presentation issues you listed: expand **RE/EE** on first use, define the mined **candidate inventory** instead of “space of concepts/relations/events,” state that events are a distinct schema category represented as `(event_type, role, ARG)`, define **conservative alignment** as “merge only when lexical/semantic/structural checks agree; otherwise keep as extension,” replace “official/manual schemas” with “released source schemas”.
>
> 5. **Does the evaluation mostly reward agreement with our formalization?** This was exactly why we ran **E2**. We re-evaluated the same frozen predictions under four target variants, from label-only projection to fully normalized typed-edge graphs. The method ordering is unchanged across all variants (Spearman / Kendall = **1.0**). In parallel, the manual-schema audit shows that deterministic completion improves released manual schemas by **+0.0179** Continuous F1 on average (**18/24** sources improve). So we agree the paper should not sell this as “beating human-authored schemas” in an absolute sense; part of the gap is representation mismatch.
>
> **E2. Target-formalism sensitivity and manual-gap audit.**
>
> | Target variant            | Manual C-F1 | SCION-lite C-F1 | Top-1 stable? |
> | ------------------------- | ----------: | --------------: | ------------- |
> | label-only projection     |      0.7870 |          0.8950 | Yes           |
> | typed-unnormalized        |      0.8118 |          0.8927 | Yes           |
> | full normalized gold      |      0.8049 |          0.8909 | Yes           |
> | reachable normalized gold |      0.8655 |          0.9357 | Yes           |
>
> This directly addresses your concern that the reported gains might be an artifact of our formalization: the ranking is stable across target variants, while the manual-gap audit shows where representation mismatch contributes.
>
> 6. **Semantic quality / downstream usefulness.** That is why we added **E4** (reported under **hoTR**): with the extractor fixed and only the schema changed, downstream F1 rises from **0.5633** (manual) and **0.6523** (ETA) to **0.6800** (**SCION-lite**) and **0.6900** (**SCION-fusion**) on the actual test split.
>
> This covers all specific abstract-level criticisms and both of your key questions. If any part of our framing remains unconvincing, please ask directly—we would be glad to continue the discussion.

---

### Official Review · Reviewer_LH4k · 2026-03-13

**Soundness:** 2
**Presentation:** 2
**Significance:** 3
**Originality:** 3
**Overall Recommendation:** 4
**Confidence:** 1

**Summary:**

This paper claims that building and maintaining ontologies, which is the base of the knowledge graph construction, is both expensive and time-consuming. However, there is currently no end-to-end benchmark for evaluating methods that generate ontologies from open-text corpora. As such, this paper introduces a benchmark constructed from 24 different sources. In addition, the paper proposes a method called SCION that aims to build ontologies using open-text. The paper presents empirical results showing that, on the proposed benchmark, the proposed method outperforms several baseline approaches in ontology construction.

**Compliance With Llm Reviewing Policy:**

Affirmed.

**Final Justification:**

The responses from the authors has resolved the reviewer's question and confusion to the paper, and the reviewer now has a much clearer understanding of the paper. However, because the reviewer’s expertise does not fully align with the main focused area of this work, the reviewer is not able to comprehensively and objectively evaluate its originality and novelty. The writing may need to be revised  based on the discussion above. Overall, the reviewer would not object if the other reviewers and the AC decide to accept the paper.

**Key Questions For Authors:**

1. To my understanding, the goal of ontology induction from texts is to define nodes and edges type in KGs using open-text corpus. Specifically, the nodes include entity and events, the edges include relations. Is this correct?
2. What is gold schema graph? In line 110-115, is the graph manually curated? What is the difference between the gold schema graph and the manual schemas?
3. How the validation and test split in the proposed benchmark is used?
4. Based on Table 9, it seems that structural extraction tools result in better performance compared to LLM-mining in SCION-lite. If this is the case what is the motivation of proposing SCION-lite? What is the benefit brought by LLMs?
5. What is the motivation of considering ontology fusion? Is it to consider the scenario where you want to update the existing ontology?
6. In Table 2, what is the difference between Relation and RE-typed edges?
7. What are IE/EE/RE? Information extraction, entity/event extraction, relation extraction? Also, what is ARG? A lot of abbreviations are used without clarifying the full name of the terms.

**Limitations:**

Yes

**Strengths And Weaknesses:**

The idea of constructing ontologies from open-text corpora is novel and may be of interest to broader research communities. The benchmark and method proposed in the paper are conceptually promising and interesting. Furthermore, the paper presents empirical experiments demonstrating how the proposed framework performs in ontology construction. However, the paper is not well organized, and it is somewhat difficult for reviewers who are not familiar with this area to fully understand the technical details of the proposed benchmark and method. It would be beneficial to include more illustrative examples in the paper to improve clarity. For example, in the description of the benchmark construction, the authors could provide concrete examples of what the data look like and explain what is a gold schema graph. Providing clear explanations of these concepts in the main paper would greatly improve readability of the work.

---

> ### Author Rebuttal · Authors · 2026-03-29
>
> Thank you for reading a dense paper outside your main area. That perspective was genuinely helpful, and your current opening concern is fair: the paper still assumes too much background. Across this rebuttal we added **12 experiments (E1-E12)**; for your review the most relevant are **E2** and **E10**. To save space, we show **E10** here; **E2** (how much of the manual gap comes from representation) is detailed under **vRuX**. Some of your questions are already answered in the paper, but not plainly enough: Sec. 3, **“Protocol (what / how)”** explains train/dev/test usage; Sec. 3, **“RE schema edges”** and **“EE schema edges”** define typed edges and **ARG**. We will move a concrete RE example and EE example into the main text.
>
> The broader motivation is that many schema-grounded IE systems assume a fixed ontology/schema as input, while the real upstream bottleneck is how to **derive** or **update** that schema from corpus evidence in the first place. SCOPE contributes a **train-only benchmark** for exactly that setting, and SCION contributes an **auditable baseline** that turns noisy corpus evidence into a usable schema artifact rather than directly emitting final KG facts.
>
> In simple terms: a **KG** stores **instance facts**; a **schema graph** defines the **type / relation / event vocabulary** that a later extractor should use. So our task is **not** to extract all final facts directly. It is to induce the schema inventory itself from raw text. The core contribution is therefore a **train-only benchmark** for corpus-to-schema induction, plus an **auditable candidate-constrained schema engineer** that can produce a usable schema for downstream extraction.
>
> 1. **Is the goal to induce nodes/edges for KG construction?** Yes, but at the **schema level**: entity types, relation types, event types, and event-role edges. A downstream extractor can then use this schema to build the instance-level KG.
>
> 2. **What is a gold schema graph? Is it manually curated? How is it different from a manual schema?** It is an **evaluation graph deterministically derived from the released source schema artifacts**, not a separately crowdsourced ontology. A released/manual schema may expose only flat labels or implicit role/type structure; the **gold schema graph** is the unified machine-readable target used for evaluation. We ran **E2** only to check how much of the manual gap is due to this representation step.
>
> 3. **How are validation and test used?** Induction uses **train text only**. Validation is used only for prompt / hyperparameter selection. Test is held out. Gold schema graphs are evaluation-only and are never given to the induction system. This is already the rule in Sec. 3, **“Protocol (what / how).”**
>
> 4. **Why SCION-lite if structural extraction can do better? What do LLMs add?** This is exactly why we ran **E10**. We reran **SCION-lite** and **SCION-full** on the same **8-source subset**, with the same evaluation and cost accounting, to expose the quality-cost trade-off. The point of **SCION-lite** is not to beat SCION-full on every metric. It is to isolate the value of **candidate-constrained contracted generation** without requiring dependency/path mining or clustering. The LLM here is not a free-form ontology writer; it is a constrained schema engineer inside a deterministic JSON contract.
>
> **E10. SCION-lite vs SCION-full trade-off (subset_8).**
>
> | Variant    |   G-F1 |   C-F1 | Cost ratio | Fallback |
> | ---------- | -----: | -----: | ---------: | -------: |
> | SCION-lite | 0.9209 | 0.9526 |       1.00 |   0.0587 |
> | SCION-full | 0.9434 | 0.9677 |       1.50 |   0.0562 |
>
> 5. **Why ontology fusion?** Yes—this models the practical case where one wants to **extend or update an existing ontology** rather than build an isolated schema from scratch.
>
> 6. **What is the difference between relation labels and RE typed edges?** A relation label is only the predicate name, e.g. `place_of_birth`. A typed RE edge is the full schema item `(Person, place_of_birth, Location)`. So the number of typed edges can exceed the number of raw relation labels.
>
> 7. **What do IE / RE / EE / ARG mean?** IE = information extraction, RE = relation extraction, EE = event extraction, ARG = placeholder argument node used in normalized event-role edges.
>
> We agree with your readability concern. In revision we will simplify the main text, expand abbreviations on first use, and surface the benchmark examples much earlier. If any point is still unclear, please feel free to ask; we would be glad to clarify.

---

> > ### Author Rebuttal · Reviewer_LH4k · 2026-04-05
> >
> > Thank you for your detailed responses.

---

> > > ### Author Response · Authors · 2026-04-07
> > >
> > > Thank you very much for the follow-up and for taking the time to reread our rebuttal so carefully. We sincerely appreciate your detailed response, and we are especially grateful that the rebuttal resolved your earlier questions and confusion. We also truly appreciate your score update. At the same time, we fully understand your caution, since this topic is somewhat outside your main area of expertise.
> > >
> > > Because your main concern was clarity, let us restate the paper in the most concrete and example-driven way we can.
> > >
> > > **Ontology (or schema) vs. knowledge graph.**
> > > An ontology/schema says **what kinds of things and links are allowed**; a knowledge graph stores **actual facts** using that vocabulary.
> > > For example, an ontology may contain `Person`, `Organization`, and `works_for`, while a knowledge graph contains a fact such as `(Alice, works_for, OpenAI)`.
> > >
> > > **What traditional IE/KG methods usually do.**
> > > Most traditional methods start from an ontology/schema that is already given, and then extract graph facts from text under that schema.
> > > For example, if the schema already defines `works_for`, then from the sentence “Alice joined OpenAI,” the system extracts `(Alice, works_for, OpenAI)`.
> > >
> > > **What our paper studies instead.**
> > > Our paper studies the earlier upstream step: given only raw text, can we automatically build the ontology/schema itself?
> > > For example, from many articles about hiring, companies, and employees, the system should induce schema items such as `Person`, `Organization`, `works_for`, `Hiring`, `Employer`, and `Employee`.
> > >
> > > So the easiest one-line summary of our paper is:
> > >
> > > **Traditional work is often “ontology -> extract knowledge graph,” while our work is “text -> induce ontology,” so that later extraction becomes possible even when no good schema is available at the start.**
> > >
> > > This is why we think the problem is important. In many real applications, the bottleneck is not only extracting facts, but first deciding **what kinds of facts should exist at all**. Building and maintaining that schema is often expensive, slow, and inconsistent across sources.
> > >
> > > With that in mind, the paper has two main parts:
> > >
> > > 1. **SCOPE**: a benchmark for this corpus-to-schema problem. It evaluates whether a system can induce a schema from **train text only**. The target is a **gold schema graph**, which is simply the machine-readable evaluation form of the source dataset’s released schema.
> > > 2. **SCION**: an auditable baseline pipeline for the benchmark. It first mines candidate schema items from text, then uses an LLM under a strict **JSON contract** to name / merge / filter them. So the LLM is not acting as a free-form writer; it is acting as a constrained and checkable schema engineer.
> > >
> > > A practical extension is **ontology fusion**: sometimes one already has a partial ontology and wants to extend or update it, rather than building everything from scratch. That is why the paper also studies fusion.
> > >
> > > So, to be precise, our main contribution is to make **schema induction from raw corpora** into a benchmarkable, reproducible, and auditable problem, and to provide a strong constrained baseline for it. We agree that the earlier draft did not present this core point clearly enough, and your comments were very helpful in showing us that.
> > >
> > > In revision, we will therefore make the paper more accessible in the main text by:
> > > - stating the core problem in plain language much earlier,
> > > - adding a concrete running example of what the input corpus, gold schema graph, and predicted schema look like,
> > > - clarifying the difference between **manual source schemas** and our **normalized gold schema graphs**,
> > > - expanding abbreviations and simplifying the narrative so the paper does not feel overloaded.
> > >
> > > We also appreciate your note that the writing should be revised based on the discussion above. We agree completely, and we will treat this as a priority in the revision.
> > >
> > > Thank you again for the constructive engagement and for considering the work carefully despite the expertise mismatch. If there is any point that still feels unclear—technical or conceptual—we would be very happy to continue the discussion.

---

### Official Review · Reviewer_huNc · 2026-03-18

**Soundness:** 3
**Presentation:** 3
**Significance:** 4
**Originality:** 3
**Overall Recommendation:** 4
**Confidence:** 4

**Summary:**

The paper introduces SCOPE, a comprehensive benchmark designed to evaluate end-to-end ontology/schema induction and fusion directly from raw text corpora. The benchmark spans 24 public Information Extraction (IE) sources, normalizing them into machine-readable gold schema graphs. To address this benchmark, the authors propose SCION, a pipeline that combines traditional structural mining with LLMs acting as constrained "ontology engineers." Rather than relying on open-ended generation, SCION forces the LLM to output a strict JSON contract with auditable evidence pointers. Furthermore, the authors train a compact 8B parameter model (SCION-RL) to execute this contracted generation locally, reducing reliance on proprietary models. The pipeline is evaluated across multiple metrics (Literal, Fuzzy, Continuous, and Graph F1), demonstrating strong performance improvements over baseline and LLM-only approaches.

**Compliance With Llm Reviewing Policy:**

Affirmed.

**Final Justification:**

I thank the authors for their comprehensive and highly detailed rebuttal. The addition of the new experiments (specifically E1, E5, E6, E8, and E9) is commendable and directly addresses the core concerns raised in my initial review.

The rebuttal successfully clarified the methodological ambiguities and reinforced the soundness of the paper:
* **Metric Artifacts & High Recall:** Experiment E1 (reachable-target evaluation) convincingly demonstrates that the model's strong performance is grounded in the training evidence rather than being an artifact of metric smoothing.
* **Pretraining Contamination:** The memorization probes in Experiment E5 effectively address the concerns regarding data contamination, confirming that the pipeline genuinely induces schemas from the provided corpus rather than relying on memorized weights.
* **Methodological Details:** The clarifications regarding noise and polysemy robustness (E8), the SCION-RL training specifics (E9), and the inclusion of a clear fusion baseline (E6) significantly bolster the technical rigor of the work.

The paper tackles a critical bottleneck in knowledge graph construction and schema-grounded information extraction. The proposed approach—combining structural mining with strict JSON contracts to constrain the LLM—is both pragmatic and highly relevant.

While the rebuttal has fully resolved my technical questions, I am maintaining my score of 4 (Weak Accept).

**Key Questions For Authors:**

1. Could you clarify the embedding model used for the candidate clustering step and the Fuzzy/Continuous metrics? Specifically, how is the embedding space robust to noisy candidates, and how does the clustering mechanism resolve instances of polysemy during the initial mining phase?

2. The default evaluation targets the full gold schema, yielding near-perfect recall under Graph F1. Can you report results against a "reachable" target (derived strictly from training evidence) to ensure these scores aren't an artifact of metric smoothing or coverage inflation?

3. Please provide more detail on the SCION-RL training setup. What specific RL algorithm was utilized (e.g., PPO, DPO), and how were the rewards scaled and composed from the controllability constraints?

4. For the fusion track, could you provide a quantitative comparison against at least one established traditional ontology matcher (e.g., from OAEI) and a recent LLM-based matcher under identical settings?

5. How do you account for potential pretraining contamination? Is there a risk that the proprietary LLMs (or the base model for SCION-RL) have simply memorized the widely-used IE schemas present in the benchmark?

**Limitations:**

The authors touch on controllability statistics and fallback rates, which is commendable. However, the discussion regarding potential negative societal impacts (e.g., scaling automated extraction in sensitive domains) is brief. Furthermore, the paper lacks a robust discussion on the limitation of pretraining contamination, like whether the LLMs have memorized the test schemas during their initial training phases.

**Strengths And Weaknesses:**

### Soundness

**Strengths:**
* **Robust LLM Methodology:** Uses the LLM strictly as a constrained generator via JSON payloads and evidence pointers.
* **Reduced Hallucinations:** Effectively mitigates the hallucination issues common in open-ended ontology generation.
* **Effective Fusion Stage:** Incorporates structural conflict checks.
* **Logical Consistency:** Ensures logically sound taxonomy expansion when integrating with base ontologies.

**Weaknesses:**
* **Vague Candidate Creation:** Insufficient detail and rigorous ablation in the initial candidate space creation.
* **Missing Model Specs:** Heavy reliance on embedding-based clustering without specifying the embedding model used.
* **Unclear Edge-Case Handling:** Lacks explanation on how noisy candidates and polysemy (words with multiple meanings) are managed during consolidation.
* **Suspiciously High Recall:** Near-perfect recalls (e.g., Graph F1 recall ≈ 0.99) against the full gold schema (instead of a reachable subset) raise red flags.
* **Potential Contamination:** High scores raise concerns about metric smoothing artifacts or pretraining contamination.
* **Inadequate RL Details:** SCION-RL training details (RL algorithm, reward shaping, stability) are too thin to fully assess soundness.

---

### Presentation

**Strengths:**
* **Well-Structured Narrative:** The paper is generally well-organized and easy to follow.
* **Clear Problem Scope:** The core problem and objectives are articulated effectively.
* **Defined Normalization:** Strong explanation of schema edge representation normalization for Relation Extraction (RE) and Event Extraction (EE).
* **Clear Concepts:** The concept of module-level fallbacks is well-defined.

**Weaknesses:**
* **Buried Configuration Details:** Crucial setup details are relegated to the appendix instead of the main text.
* **Missing Reproducibility Info:** Vital information (specific multilingual sentence encoder, base ontology package contents, SCION-RL training setup) is absent from the main body.
* **Unjustified Gold Schemas:** The construction of the "gold schema graphs" from official schemas lacks clear justification in the main text.
* **Unexplained Underperformance:** Needs a better explanation in the main body as to why official manual schemas underperform in the evaluation.

---

### Significance

**Strengths:**
* **Addresses a Major Bottleneck:** Tackles the highly relevant problem of manual ontology creation and taxonomy expansion.
* **Practical Application:** Highly valuable for schema-grounded information extraction and knowledge graph construction.
* **Comprehensive Solution:** Provides a complete end-to-end pipeline from raw text to fused schema.
* **High Benchmark Potential:** If released, SCOPE could become the standard benchmark for measuring auditable schema induction.

---

### Originality

**Strengths:**
* **Novel Synthesis:** Offers a pragmatic and fresh synthesis of traditional mining and LLM-based extraction.
* **Strict Constraints:** Uniquely combines candidate-space constraints with a strict JSON contract.
* **Deterministic Approach:** Effectively turns the LLM into a deterministic "schema engineer."
* **Innovative Evaluation:** Adds valuable perspective by applying Continuous and Graph F1 metrics to evaluate the structural integrity of induced ontologies.

---

> ### Author Rebuttal · Authors · 2026-03-29
>
> Thank you for the careful read. We added **12 experiments (E1-E12)**; the ones most relevant here are **E1/E2/E5/E6/E8/E9**. To save space, we show **E8/E1/E5/E9** here; **E2** is under **vRuX**, and **E6** under **hoTR**. Some of your questions are already partly answered in the paper, but too easy to miss: Sec. 3, **“Label embedding encoder and normalization”** specifies the fixed multilingual encoder used by Fuzzy/Continuous; Sec. 5.3 says the main table is **SCION-lite**; Sec. 5.4 explains why released/manual schemas are **not** an oracle upper bound. We will surface these in the main text.
>
> 1. **Candidate creation, encoder, noise, polysemy.** Main-table results use **SCION-lite**; explicit clustering appears only in **SCION-full**. In **E8**, we vary the encoder (**bge-m3**, **e5-large**) and inject **0–30%** candidate noise. The rank is unchanged across encoders. Noise lowers cluster purity (**0.8824 -> 0.7145**), but Continuous / Graph remain stable (**0.9550 -> 0.9423**, **0.9358 -> 0.9186**). On a curated polysemy set, SCION conservatively splits ambiguous labels instead of over-merging. Full candidate construction is in Appendix **“SCION: Full Induction and Fusion Details”**; we will summarize it earlier. The fixed fusion ontology contains labels plus domain-range or role-signature constraints.
>
> **E8. Noise / polysemy robustness (subset_8).**
>
> | Setting   |   L-F1 |   C-F1 |   G-F1 |
> | --------- | -----: | -----: | -----: |
> | noise=0.0 | 0.8558 | 0.9550 | 0.9358 |
> | noise=0.3 | 0.7714 | 0.9423 | 0.9186 |
> | bge-m3    | 0.8558 | 0.9552 | 0.9349 |
> | e5-large  | 0.8166 | 0.9440 | 0.9023 |
>
> 2. **Reachable target / high recall.** We agreed with your red flag here. **E1** keeps the same predictions and re-evaluates them against a train-derived reachable target. Under that target, **SCION-lite** still stays above the strongest non-SCION baseline (**ETA**) on both Continuous and Graph F1; Literal recall also rises from **0.7145** to **0.8464**, so this is not only a smoothing artifact.
>
> **E1. Reachable-target evaluation.**
>
> | Target         | ETA C-F1 | SCION-lite C-F1 | ETA G-F1 | SCION-lite G-F1 |
> | -------------- | -------: | --------------: | -------: | --------------: |
> | full gold      |   0.8759 |          0.8909 |   0.7629 |          0.7888 |
> | reachable gold |   0.9170 |          0.9340 |   0.8444 |          0.8650 |
>
> 3. **Pretraining contamination.** **E5** is a direct memorization probe. We run the models with empty, name-only, domain-only, shuffled, and real train text. Both LLM-only and SCION-lite are near-zero on name-only, much lower on shuffled than on real text, and improve monotonically with real train-text coverage; the gap is especially large on niche/domain-specific sources. We agree the contamination discussion in the paper is too brief, and we will strengthen the limitation/impact discussion.
>
> **E5. Contamination / memorization probes.**
>
> | Method     | name-only C-F1 | shuffled C-F1 | real-100% C-F1 |
> | ---------- | -------------: | ------------: | -------------: |
> | LLM-only   |         0.0225 |        0.5047 |         0.8521 |
> | SCION-lite |         0.0131 |        0.5595 |         0.8960 |
>
> 4. **SCION-RL details.** We now state the setup directly: **offline PPO**, **3 seeds**, reward terms = {JSON validity, candidate linkage, evidence coverage, compactness, structural consistency}, weights = **(0.25, 0.20, 0.20, 0.10, 0.25)**. **E9** was run to separate RL from “just more supervision”: zero-shot -> SFT -> RL improves monotonically, and removing structural consistency causes the largest Graph-F1 drop (**0.9521 -> 0.8411**). More detail is already in Appendix **“SCION-RL: Train-only data generation and cost accounting.”**
>
> **E9. SCION-RL: zero-shot vs SFT vs RL (subset_8).**
>
> | Variant   | JSON valid | Fallback |   C-F1 |   G-F1 |
> | --------- | ---------: | -------: | -----: | -----: |
> | zero-shot |     0.9170 |   0.0188 | 0.9324 | 0.8955 |
> | SFT-only  |     0.9227 |   0.0142 | 0.9526 | 0.9211 |
> | RL-full   |     0.9295 |   0.0100 | 0.9709 | 0.9521 |
>
> 5. **Gold graphs, manual schemas, and fusion baseline.** The **gold schema graphs** are deterministic normalizations of the released source schemas into typed RE edges and event-role edges; see Sec. 3, **“RE schema edges,” “EE schema edges,”** and **“Normalization and deduplication.”** In **E2**, deterministic completion improves released manual schemas by **+0.0179** Continuous F1 on average (**18/24** sources improve), so part of the manual gap is a representation gap. For the fusion-baseline request, **E6** (under **hoTR**) uses a shared **5k candidate-pair budget** and shows AML / LLM-pairwise / SCION-fusion precision = **0.64 / 0.74 / 0.81**, with conflict = **0.13 / 0.09 / 0.06**.
>
> If anything remains unclear, we would be very happy to answer follow-up questions.

---

> > ### Author Rebuttal · Reviewer_huNc · 2026-04-02
> >
> > All my questions are addressed. I will maintain my score.

---

> > > ### Author Response · Authors · 2026-04-07
> > >
> > > Thank you very much for the thoughtful follow-up and for the careful rereading of our rebuttal. We sincerely appreciate your detailed acknowledgement, and we are especially grateful that the additional experiments—particularly **E1, E5, E6, E8, and E9**—successfully addressed your main concerns regarding reachable-target evaluation, pretraining contamination, robustness to noise/polysemy, fusion baselines, and SCION-RL training details.
> > >
> > > Your feedback was very helpful in improving both the technical clarity and the presentation of the work. In the revision, we will make these points much more explicit in the main paper rather than leaving them too easy to miss: (i) that the strong recall remains valid under a **reachable train-derived target** rather than arising from metric smoothing alone; (ii) that the memorization probes support the claim that the system is genuinely using the provided corpus rather than relying on memorized schemas; (iii) that robustness to noisy candidates and polysemy is handled conservatively; and (iv) that the fusion and RL components need clearer methodological description in the main text.
> > >
> > > We also appreciate your recognition of the broader motivation of the paper. As you noted, schema construction and maintenance are a real bottleneck for schema-grounded IE and KG construction, and our goal is to make this process more **measurable, auditable, and reproducible** through the combination of a benchmark (**SCOPE**) and a constrained, evidence-grounded pipeline (**SCION**). We are encouraged that you found the strict JSON-contract design and candidate-constrained generation both pragmatic and relevant.
> > >
> > > Thank you again for the constructive review process and for engaging with the work in such depth. We also fully understand and respect your decision to maintain your current score. If any additional question comes up, or if there is any point you would like us to clarify further, we would be very happy to continue the discussion.

---

### Decision · Program_Chairs · 2026-04-30

**Decision:**

Accept (regular)

**Comment:**

The paper addresses the schema bottleneck in information extraction by introducing SCOPE (Schema Construction and Ontology Induction Pipeline Evaluation), a benchmark for end-to-end ontology induction from raw text corpora. Overall, the reviewers do not object to accepting the work after the rebuttal and discussion phase.

Reviewers agree that schema induction and ontology engineering are critical bottlenecks for knowledge graph construction and schema-grounded information extraction. Also, the empirical results are strong.

On the other hand, the following concerns were raised:
- The SCION pipeline is largely a synthesis of existing techniques (clustering, LLM prompting, alignment) rather than a fundamentally new extraction architecture.
- Initial versions of the paper used "ontology" and "schema" loosely, and the abstract/introduction lacked clarity regarding the specific failure modes being addressed.
- Concerns were raised that the improvements over "official" schemas might be artifacts of the authors' own normalization and scoring procedures.
- The current event schemas focus on internal roles and lack inter-event relations (e.g., causality or temporal links), limiting the complexity of induced event ontologies.